# Efficacy of morning versus afternoon aerobic exercise training on reducing metabolic syndrome components: A randomized controlled trial

Felix Morales-Palomo[1] , Alfonso Moreno-Cabañas[1,2,3] , Laura Alvarez-Jimenez[1] , Diego Mora-Gonzalez[4], Juan F. Ortega[1]  and Ricardo Mora-Rodriguez[1]

[1]*Exercise Physiology Lab at Toledo, University of Castilla-La Mancha, Toledo, Spain*
[2]*Centre for Nutrition, Exercise, and Metabolism, University of Bath, Bath, UK*
[3]*Department for Health, University of Bath, Bath, UK*
[4]*Department of Nursing, Physiotherapy and Occupational therapy, University of Castilla-La Mancha, Toledo, Spain*

Handling Editors: Paul Greenhaff & Bettina Mittendorfer

The peer review history is available in the Supporting Information section of this article (https://doi.org/10.1113/JP285366#support-information-section).

The Journal of **Physiology**

**Abstract**  A supervised intense aerobic exercise program improves the health of individuals with metabolic syndrome (MetS). However, it is unclear whether the timing of training within the 24 h

Clinicaltrials.gov identifier NCT04477590.

The Journal of Physiology

day would influence those health benefits. The present study aimed to determine the influence of morning *vs.* afternoon exercise on body composition, cardiometabolic health and components of MetS. One hundred thirty-nine individuals with MetS were block randomized into morning (AMEX; $n = 42$) or afternoon (PMEX; $n = 59$) exercise training groups, or a non-training control group (Control; $n = 38$). Exercise training was comprised of 48 supervised high-intensity interval sessions distributed over 16 weeks. Body composition, cardiorespiratory fitness (assessed by $\dot{V}_{O_2max}$), maximal fat oxidation ($FO_{max}$), blood pressure and blood metabolites were assessed before and after the intervention. Compared with the non-training Control, both exercise groups improved similarly body composition (–0.7% fat loss; $P = 0.002$), waist circumference (–2.1 cm; $P < 0.001$), diastolic blood pressure (–3.8 mmHg; $P = 0.004$) and $\dot{V}_{O_2max}$ (3.5 mL kg$^{-1}$ min$^{-1}$; $P < 0.001$) with no differences between training groups. AMEX, in comparison with PMEX, reduced systolic blood pressure (–4% *vs.* –1%; $P = 0.019$), plasma fasting insulin concentration (–12% *vs.* –5%; $P = 0.001$) and insulin resistance (–14% *vs.* –4%; $P = 0.006$). Furthermore, MetS $Z$ score was further reduced in the AMEX compared to PMEX (–52% *vs.* –19%; $P = 0.021$) after training. In summary, high-intensity aerobic exercise training in the morning in comparison to training in the afternoon is somewhat more efficient at reducing cardiometabolic risk factors (i.e. systolic blood pressure and insulin sensitivity).

(Received 25 July 2023; accepted after revision 16 October 2023; first published online 31 October 2023)

**Corresponding author** R. Mora-Rodriguez: Exercise Physiology Lab at Toledo, University of Castilla-La Mancha, 45071 Toledo, Spain. Email: ricardo.mora@uclm.es

**Abstract figure legend** Exercise training can be used as a non-pharmacological treatment to halt, or at least reduce, the progression of the cardiovascular and metabolic derangements that compose the metabolic syndrome. Recent studies using intense interval training have spurred interest in other exercise factors that may increase exercise training effectiveness. For example, the time of day at which training takes place could modulate the health outcome. The present study randomly assigned 175 individuals with metabolic syndrome to train in the morning or afternoon, or to remain untrained. The novel finding was that working out in the morning further improved, systolic blood pressure, insulin resistance and, in general, the metabolic syndrome (i.e. $Z$ score) compared to training in the afternoon. Thus, morning training is somewhat more effective in maximizing health promotion in individuals with metabolic syndrome.

### Key points

- The effect of exercise time of day on health promotion is an area that has gained interest in recent years; however, large-scale, randomized-control studies are scarce.
- People with metabolic syndrome (MetS) are at risk of developing cardiometabolic diseases and reductions in this risk with exercise training can be precisely gauged using a compound score sensitive to subtle evolution in each MetS component (i.e. $Z$ score).
- Supervised aerobic exercise for 16 weeks (morning and afternoon), without dietary restriction, improved cardiorespiratory and metabolic fitness, body composition and mean arterial pressure compared to a non-exercise control group.
- However, training in the morning, without changes in exercise dose or intensity, reduced systolic blood pressure and insulin resistance further compared to when training in the afternoon.
- Thus, high-intensity aerobic exercise training in the morning is somewhat more efficient in improving the health of individuals with metabolic syndrome.

## Introduction

Metabolic syndrome (MetS) is a cluster of cardiometabolic risk factors and comorbidities conveying a high risk of cardiovascular disease and type 2 diabetes (Alberti et al., 2009). This syndrome has become a major public health threat in the US and worldwide since an estimated 33% of individuals between the ages of 20 and 60 have MetS (Aguilar et al., 2015). Lifestyle interventions based on supervised exercise are cost-effective strategies to halt the progression in MetS that occurs with aging (Morales-Palomo et al., 2023). The impact of exercise

training on health improvements depends mainly on the frequency, intensity, time, and type of exercise (FITT (Barisic et al., 2011)). However, genetics (Bouchard et al., 2011) or even the timing of exercise concerning meal ingestion (Edinburgh et al., 2019) could modulate the training adaptations that promote health.

Circadian rhythms are biological responses that follow a 24-hour cycle and respond primarily to light and dark phases (Gabriel & Zierath, 2022). Circadian rhythms affect several physiological processes influencing appetite, sleep/wake cycles, and even exercise performance (Chaput et al., 2023; Gabriel & Zierath, 2019; Mora-Rodríguez et al., 2012). Some authors have suggested that the desynchronization of natural circadian rhythms is at the core of the etiology of the MetS (Maury et al., 2010; Zimmet et al., 2019). Conversely, adequate timing of exercise could promote phase adjustment of circadian rhythms. Therefore, exercise used as a Zeitgeber (i.e. circadian rhythm cue) could help to fight cardiometabolic disease and MetS (Gabriel & Zierath, 2019). Timing exercise bouts to coordinate with an individual's circadian rhythms might be an efficacious strategy to optimize the health benefits of exercise.

The effect of exercise time of day on health promotion is an area that has gained research interest in recent years. In the past, the time of day at which exercise was prescribed or undertaken was often unreported in protocol descriptions or not controlled. Despite recent experiments devoted to this topic, the timing of exercise remains a controversial issue, with some studies favouring evening exercise to improve glycaemic control (Savikj et al., 2019), metabolic benefits (van der Velde et al., 2022), weight loss (Di Blasio et al., 2010), fat mass loss or exercise performance (Mancilla et al., 2021). By contrast, several studies report that exercise in the morning may be associated with more effective management of obesity (Alizadeh et al., 2017; Chomistek et al., 2016; Schumacher et al., 2020; Willis et al., 2020) and lower risk of incident cardiovascular diseases (Albalak et al., 2022). Lastly, some studies argue for a lack of differential effect of exercise timing (Janssen et al., 2022) in glycaemic control (Teo et al., 2020), weight loss (Blankenship et al., 2021; Brooker et al., 2023) or cardiorespiratory fitness (CRF) levels (i.e. $\dot{V}_{O_2max}$) (Brooker et al., 2023; Mancilla et al., 2021).

An intense supervised aerobic exercise program (8–20 weeks in duration) has repeatedly been shown to improve health parameters in individuals with cardio-metabolic diseases (Johnson et al., 2007; Morales-Palomo et al., 2019). However, it is unclear whether the timing of training within the 24 h day would influence those health benefits in patients with the metabolic syndrome (MetS). The present study aimed to determine whether there are differences in the therapeutic impact of a 16 week long high-intensity aerobic exercise program on MetS Z score (primary outcome), performed either in the morning or afternoon. We assessed MetS progression using a sensitive MetS Z score, cardiorespiratory ($\dot{V}_{O_2max}$) and metabolic (maximal fat oxidation rates) fitness, muscle power, body composition, and blood markers of insulin resistance and dyslipidaemia.

## Methods

### Experimental design and participants

In total, 139 volunteers (49 women and 90 men; body mass index, $30.6 \pm 3.0$ kg m$^{-2}$) fulfilling MetS criteria as per harmonized definition (Alberti et al., 2009) using population Europid waist circumference cutpoints completed the study. MetS was defined as the presence of three of the following five risk factors; elevated waist circumference ($\geq 94$ cm for males and $\geq 80$ cm for females), elevated blood pressure ($>130$ mmHg for systolic and/or $>85$ mmHg for diastolic), elevated fasting blood glucose ($\geq 100$ mg dL$^{-1}$), triglycerides (i.e. $\geq 150$ mg dL$^{-1}$) and reduced high-density lipoprotein (HDL)-cholesterol (i.e. $<40$ mg dL$^{-1}$ for males and $<50$ mg dL$^{-1}$ for females). Taking a prescription for any of the MetS components was accounted for comparable to the presence of the factor. Participants were previously inactive ($<150$ min week$^{-1}$ of the moderate-intensity activity assessed by 7 day International physical activity questionnaire) (Craig et al., 2003). Subjects under medical treatment were instructed to continue with their current medication prescriptions during the study (Table 1). Exclusion criteria included untreated cardiovascular or renal disease, peripheral vascular disease and any disease

**Felix Morales Palomo**, assistant professor and scientist at the Exercise Physiology Lab at the University of Castilla-La Mancha. His main research interest is to determine the dose and type of exercise that more efficiently improves the health of people with cardiometabolic diseases. **Ricardo Mora-Rodriguez**, directs the Exercise Physiology Lab at the University of Castilla-La Mancha in Spain. During the last decade, his research has focused on understanding how exercise, medication and diet improve the health of obese-metabolic syndrome individuals.

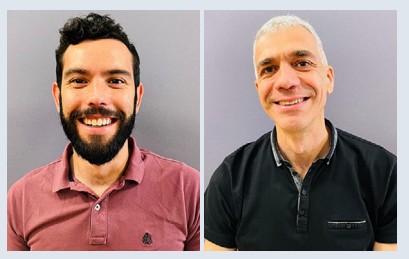

**Table 1. Participants' characteristics before the intervention**

|  | AMEX (*n* = 42) | PMEX (*n* = 59) | Control (*n* = 38) | *P* value |
|---|---|---|---|---|
| Age | 57 ± 7 | 56 ± 7 | 60 ± 7 | 0.28 |
| % Women | 38% | 36% | 32% | 0.17 |
| MetS components | 3.2 ± 0.9 | 3.4 ± 1.0 | 3.4 ± 1.0 | 0.54 |
| Body weight (kg) | 82.2 ± 11.4 | 85.7 ± 12.9 | 87.3 ± 11.1 | 0.15 |
| Body fat (kg) | 27.8 ± 5.4 | 28.7 ± 5.8 | 29.8 ± 6.4 | 0.34 |
| Fat-free mass (kg) | 54.4 ± 10.0 | 57.0 ± 11.2 | 57.5 ± 10.7 | 0.36 |
| $\dot{V}_{O_2max}$ (mL kg$^{-1}$ min$^{-1}$) | 23.3 ± 5.0 | 24.4 ± 5.7 | 24.2 ± 5.9 | 0.60 |
| $\dot{V}_{O_2max}$ (mL kg FFM$^{-1}$ min$^{-1}$) | 35.2 ± 6.4 | 36.6 ± 6.9 | 36.6 ± 6.3 | 0.55 |
| **Medication (user in %)** | | | | ***P* value of Chi-squared** |
| Anti-hypertensive | 52% | 63% | 61% | 0.57 |
| Cholesterol-lowering | 48% | 46% | 47% | 0.98 |
| Triglyceride-lowering | 0% | 5% | 11% | 0.29 |
| Glucose-lowering | 19% | 19% | 21% | 0.96 |
| No medication | 15% | 33% | 16% | 0.59 |
| One medication | 44% | 29% | 45% | 0.22 |
| Two medications | 36% | 26% | 21% | 0.28 |
| Three medications | 5% | 10% | 16% | 0.21 |
| Four medications | 0% | 2% | 3% | 0.47 |

Data presented as the mean ± SD, FFM, fat-free mass.

associated with exercise intolerance. Body weight stability in the 4 months before study enrolment was an additional requirement (gain or loss <4 kg). All subjects provided written, witnessed and informed consent under a protocol approved by the local Virgen de la Salud Hospital's Ethics Committee and in accordance with the *Declaration of Helsinki*. This is a substudy part of a larger clinical trial evaluating the interactions between the timing of oral medicines, exercise training and meal intake in the evolution of the factors that compose the MetS (ClinicalTrials.gov Identifier: NCT04477590).

Volunteers were recruited, clinically screened, randomized and tested as depicted in Fig. 1 in compliance with CONSORT (Consolidated Standards of Reporting Trials). Following baseline testing, participants were randomized to one of three groups: morning exercise

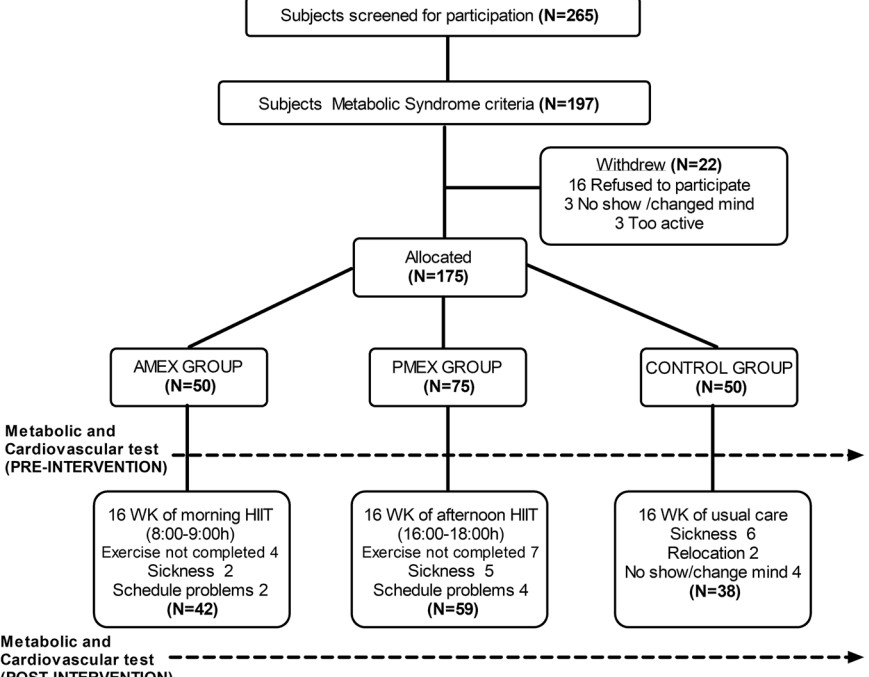

**Figure 1. CONSORT schematic**
CONSORT (Consolidated Standards of Reporting Trials) schematic representation of the study procedures.

(AMEX), afternoon exercise (PMEX) or waitlist control (Control) at a 1:1.5:1 (AMEX:PMEX:Control) ratio using a block randomized (by sex, age and body mass index) controlled design. We increased participants in the PMEX group to represent that more individuals in our country exercise after the working day shift (typically between 07.00 h and 15.00 h) and the ingestion of the main meal. Participants allocated to Control were asked to continue with their usual day-to-day activities and were offered the exercise program after the study was completed.

### Intervention

Exercise groups followed 16 weeks of a supervised aerobic training program with a frequency of three times per week (Monday, Wednesday and Friday) consisting of high-intensity interval training on stationary bikes (Tomahawk S-Serie; Indoorcycling Group GmbH, Nürnberg, Germany). Participants were required to complete at least 90% of the exercise sessions. Our dietitian had individual interviews to advise a meal of common foodstuff with a target macronutrient distribution range for carbohydrates (45–65% of energy), protein (10–35% of energy) and fat (20–35% of energy; limited saturated and trans fats) (Manore, 2005). All subjects were advised to maintain their normal dietary and physical activity habits during the whole study. Monthly, during the intervention period (16 weeks), subjects filled out a 3 day nutritional diary that was analysed for caloric intake and macronutrient composition with software that included common Spanish foodstuff (University of Barcelona; CESNID, version 1.0; Barcelona, Spain). Similarly, every month, subjects wore a wristband activity monitor (Polar Electro, Kempele, Finland) for 48 h to monitor steps per day, as well as standing and supine time. Participants in both the AMEX and the PMEX groups completed their training sessions between 08.00 and 9.00 h and 16.00 and 18.00 h, respectively. Participants were required to consume a meal at least 1 h before the start of each training session. Each training session included a 10-min warm-up at 70% of maximum heart rate ($HR_{max}$) followed by $4 \times 4$ min intervals at 90% of $HR_{max}$ interspersed with a 3 min active recovery at 70% of $HR_{max}$ and a 5 min cool-down period. Exercise intensity was prescribed as a percentage of the maximum heart rate (%$HR_{max}$) achieved during the pre-intervention maximal exercise test. During all training sessions, heart rate was continuously displayed on a large screen and participants adjusted the workload to comply with their personalized heart rate training zone (Seego Realtrack Sytems, Almeria, Spain) under the supervision of one member of the research team. We followed the progression principle of training and during the first 3 weeks (nine training sessions) the hard bouts were expanded from 2

to 4 min in duration and from two bouts to four bouts in number. Monthly, during a maximal cycling bout in a regular training session, $HR_{max}$ was re-evaluated and training workloads were adjusted accordingly. Mid-training (week 8) all subjects in each exercising group were studied in the laboratory during a regular exercise high-intensity interval session using an electronically braked cycle ergometer (Ergoselect 200; Ergoline, Windhagen, Germany). $\dot{V}_{O_2}$, $\dot{V}_{CO_2}$, heart rate and power output were collected continuously during the whole exercise session. Exhaled air was continuously collected and analysed breath-by-breath for oxygen consumption and carbon dioxide production using indirect calorimetry (Quark b2; Cosmed, Rome, Italy). $\dot{V}_{O_2}$ and $\dot{V}_{CO_2}$ were used to calculate energy expenditure as proposed by Brouwer (1957).

### Outcomes

**MetS components and other health indicators.** Before and after the 4 months of intervention patients arrived at the laboratory in the morning after a 10–12 h overnight fast. For the training groups, post-training measurements were scheduled at least 48 h after the last exercise training session to examine the chronic effects of exercise training rather than the acute most recent exercise session. Nude body weight (Hawk; Metler, Toledo, OH, USA), height (Stadiometer Secca 217; Secca, Hamburg, Germany), and waist circumference were measured using a non-elastic measuring tape. Fat mass and fat-free mass were determined by bioelectrical impedance analysis (Tanita BC-418; Tanita Corp, Tokyo, Japan). After 10 min of supine rest, blood pressure was measured in triplicate using a calibrated ECG-gated electro-sphygmomanometer (Tango; Suntec Medical, Durham, NC, USA).

Thereafter, on a different day, within the same week, a 7 mL blood sample was collected to determine serum glucose, insulin and lipid levels [triglycerides, total cholesterol, HDL-cholesterol and low-density lipoprotein (LDL)-cholesterol]. Insulin sensitivity was calculated using the homeostasis model assessment for insulin resistance (HOMA-IR) (Matthews et al., 1985). Sex-specific $Z$ scores were calculated for each MetS criterion using the group SD, with the sum of the $Z$ scores for each MetS component divided by six to compile the MetS risk score with units of SD (Mora-Rodriguez et al., 2016). The equations used to calculate the MetS $Z$-score were:

$$\text{Men's MetS } Z \text{ score} = [(40 - HDL - cholesterol)/SD]$$

$$+ [(triglycerides - 150)/SD]$$

$$+ [(glucose - 100)SD]$$

$$+ [(waist circumference - 94)/SD]$$

$$+ [(systolic blood pressure - 130)$$

**Table 2. Characteristics of training bouts**

| | AMEX ($n = 42$) | PMEX ($n = 59$) | Control ($n = 38$) | *P* value |
|---|---|---|---|---|
| Training sessions compliance | 92% | 90% | – | – |
| **Exercise intensity** | | | | |
| Average heart rate (beats $min^{-1}$) | $130 \pm 12$ | $133 \pm 15$ | – | 0.23 |
| Average heart rate at 70% (beats $min^{-1}$) | $116 \pm 13$ | $119 \pm 15$ | – | 0.37 |
| Average heart rate at 90% (beats $min^{-1}$) | $145 \pm 15$ | $147 \pm 20$ | – | 0.28 |
| Average heart rate (%$HR_{max}$) | 79% | 81% | – | 0.56 |
| Average heart rate at 70% (%$HR_{max}$) | 71% | 72% | – | 0.88 |
| Average heart rate at 90% (%$HR_{max}$) | 90% | 92% | – | 0.47 |
| Average workload (W) | $114 \pm 29$ | $118 \pm 26$ | – | 0.25 |
| Average workload (%$W_{max}$) | 59% | 56% | – | 0.43 |
| Average $\dot{V}O_2$ (L $min^{-1}$) | $1.33 \pm 0.39$ | $1.41 \pm 0.42$ | – | 0.36 |
| Average $\dot{V}O_2$ (%$\dot{V}_{O_2 max}$) | 69% | 67% | – | 0.42 |
| Average energy expenditure (kcal) | $383 \pm 80$ | $393 \pm 85$ | – | 0.52 |

Data presented as the mean $\pm$ SD.

$$/SD] + [(\text{diastolic blood pressure} - 85)/SD] \quad (1)$$

$$\text{Women's MetS } Z \text{ score} = [(50 - HDL - \text{cholesterol})/SD]$$
$$+ [(\text{triglycerides} - 150)/SD]$$
$$+ [(\text{glucose} - 100)SD]$$
$$+ [(\text{waist circumference} - 80)$$
$$/ SD] + [(\text{systolic blood pressure} - 130)/SD] + [(\text{diastolic blood}$$
$$\text{pressure} - 85)/SD] \quad (2)$$

**Metabolic and cardiorespiratory fitness.** Maximal fat oxidation rate ($FO_{max}$) was assessed during a submaximal graded exercise test in a fasted state and using an electronically braked cycle ergometer (Ergoselect 200; Ergoline). The initial workload was set at 10 W for women and 30 W for men with increases of 10 W and 15 W each 4 min, respectively. Exhaled air was continuously collected and analysed breath-by-breath for oxygen consumption and carbon dioxide production using indirect calorimetry (Quark b2; Cosmed). The trial was terminated when the respiratory exchange ratio (RER = $\dot{V}_{CO_2}/\dot{V}_{O_2}$) exceeded 1.0. The last minute of each stage was averaged to calculate the non-protein respiratory quotient and fat oxidation rate (Péronnet & Massicotte, 1991). Exercise stages were extended from the standard 3 min (Achten et al., 2002) to 4 min duration to gain sensitivity and accurately detect $FO_{max}$ in our highly deconditioned patients. After 15 min of passive rest and rehydration (200 mL of orange juice containing 20 g of carbohydrate), maximal oxygen uptake ($\dot{V}_{O_2 max}$) and maximal power output ($W_{max}$) were assessed on an electronically braked cycle ergometer (Ergoselect 200; Ergoline) during a graded exercise test using indirect calorimetry (Quark b2; Cosmed). After 3 min of warm-up at 30 W for women and 50 W for men, the workload increased every minute (15 W for women and 20 W for men) until volitional exhaustion. This test was followed by a verification test at 110% of the maximal workload reached to ensure the achievement of actual $\dot{V}_{O_2 max}$ (Moreno-Cabanas et al., 2020).

## Statistical analysis

We used per-protocol analysis and only patients who completed the intervention were included in the statistical analysis. Sample size calculation was based on changes in MetS $Z$ score data (main outcome) in individuals with MetS completing 4–6 months of aerobic exercise training program similar to the one in the present study (Earnest et al., 2013; Johnson et al., 2007; Morales-Palomo et al., 2019; Mora-Rodriguez et al., 2016, 2018). Assuming a power of 80% and a $\alpha$-error probability of 0.05, it was calculated that 22 patients were required to detect a significant effect of exercise training on improving MetS $Z$ score. Because the attrition rates in exercise interventions could reach 20% (Groeneveld et al., 2009) and because it is uncertain what the differences would be between training in the AM *vs.* PM, we doubled the calculated required sample size to a target sample of at least 50 participants in each group. Data are presented as the mean $\pm$ SD. 95% confidence intervals (CI) were also calculated. A Smirnov–Kolmogorov test revealed that data were normally distributed. Between-group

comparisons at baseline were performed via one-way analysis of variance (Table 1) and a Student's *t* test for independent samples (Table 2). When categorical variables were analysed, a chi-squared test was performed. (Table 1). Mixed-design (split-plot) analysis of covariance was run to analyse differences across time (baseline *vs.* 16 weeks of training) and between experimental groups in all reported variables, adjusted by baseline values. This design tested the differences between the three groups (AMEX, PMEX and Control) when participants underwent repeated measures (PRE and POST) in the primary and secondary outcome measures. To minimize the risk of statistical type I error, *post hoc* pairwise comparisons (Tukey's test) were only performed between groups when a significant time × group interaction was found. To improve the interpretation of the differences, the effect size of time and time × group interaction were calculated using eta squared ($\eta^2$). The effect size obtained from $\eta^2$ was considered large if ≥0.14, moderate if ≥0.06, and small if <0.06. SPSS, version 28 (IBM Corp., Armonk, NY, USA) was used for statistical analysis. $P < 0.05$ was considered statistically significant.

## Results

### Subjects and exercise characteristics

Participants were all white living in southern Europe. Women participants comprised 35% of all subjects. Data were analysed without sex differentiation because all women were postmenopausal and were not taking hormonal replacement (Guio de Prada et al., 2019) and their responses did not differ from men's responses in primary and secondary outcomes of the study [time × sex interaction; MetS *Z* score, $P = 0.20$; body weight (kg), $P = 0.06$; $\dot{V}_{O_2max}$ (L min$^{-1}$), $P = 0.70$]. No differences existed between groups at baseline in the number of women, age, body composition, number of MetS components, $\dot{V}_{O_2max}$ or medicine use (Table 1). There were also no changes in the use of medications to treat MetS after the intervention in any group ($P = 0.37$ for time × group) (Table 3). There were no significant differences between exercise groups for training compliance, exercise training intensity, and exercise-induced energy expenditure per session (all $P > 0.05$) (Table 2). The withdrawal rate was similar in all three groups (AMEX,16%; PMEX; 21%; and Control, 24%; *P* value of $\chi^2 = 0.59$) and always took place during the first 3 weeks of the experiment. Discontinuation in study participation was a result of changes in work schedule, unable to attend to the post-16 week test and minor injuries or diseases (back pain and prolonged flu-like symptoms) (Fig. 1).

### Body weight and composition

The changes in body weight and body composition following the intervention are depicted in Table 3. After 16 weeks, there was a significant time effect in fat-free mass ($P = 0.037$); however, there were no significant time × group interaction effects between groups ($P = 0.13$). A significant time effect ($P = 0.008$) and time × group interaction ($P = 0.002$) was found after 16 weeks of training in fat mass with reductions in both exercise groups (AMEX: −0.8 kg; 95% CI = −1.45 to −0.21, $P = 0.009$; PMEX: −0.6 kg; 95% CI = −1.17 to −0.13, $P = 0.015$) and increase in the CONTROL group (0.6 kg; 95% CI = 0.03–1.37, $P = 0.04$. After the intervention, the *post hoc* analysis showed differences in fat mass in both exercise groups from the Control group (AMEX, $P = 0.004$; PMEX, $P = 0.006$).

### Metabolic and cardiorespiratory fitness

CRF (i.e. $\dot{V}_{O_2max}$), $W_{max}$ and $FO_{max}$ after 16 weeks are shown in Table 3. Before intervention $\dot{V}_{O_2max}$, $W_{max}$ and $FO_{max}$ were similar among the groups. A significant time × group interaction effect emerged for $\dot{V}_{O_2max}$ ($P < 0.001$) and $W_{max}$ ($P < 0.001$). After 16 weeks, the Control group had a slight but no significant reduction in $\dot{V}_{O_2max}$ (−3%; 95% CI = −0.15 to 0.02 L min$^{-1}$, $P = 0.11$), whereas both exercise groups improved their $\dot{V}_{O_2max}$ (L min$^{-1}$; AMEX: 16%; 95% CI = 0.23–0.39; PMEX: 11%; 95% CI = 0.16–0.30; both $P < 0.001$). Because body weight did not significantly change in any group (Table 3), $\dot{V}_{O_2max}$ per kilogram of body weight (i.e. mL kg$^{-1}$ min$^{-1}$) was similar to the absolute values (i.e. LO$_2$ min$^{-1}$) in the exercise groups. However, the Control group reduced their $\dot{V}_{O_2max}$ per kilogram of body weight (mL kg$^{-1}$ min$^{-1}$; −5%; 95% CI = −2.10 to −0.06, $P = 0.039$). The increase in CRF in both exercise groups was paralleled by enhanced maximal power output ($W_{max}$; AMEX: 21%; 95% CI = 27–39 W; PMEX: 15%; 95% CI = 21–31 W; both $P < 0.001$) with no differences between exercise groups ($P > 0.05$). A significant time × group interaction effect was found for $FO_{max}$ ($P = 0.01$). Although, in the Control group, there were no changes, after 16 weeks, both exercise groups improved $FO_{max}$ similarly (AMEX: 0.05 g min$^{-1}$; 95% CI = 0.03–0.07; PMEX: 0.06 g min$^{-1}$; 95% CI = 0.04–0.08; both $P < 0.001$). At the end of the intervention, the improvements in $\dot{V}_{O_2max}$, $W_{max}$ and $FO_{max}$ in both exercise groups (AMEX and PMEX) were larger than in Control after conducting *post hoc* analyses (all $P < 0.05$).

### MetS components and *Z* score

The evolution of MetS components after 16 weeks of intervention is depicted in Table 3. Before the intervention,

**Table 3. Anthropometric, metabolic syndrome factors, and clinical variables before and after 16 weeks of high-intensity interval training**

| | AMEX (n = 42) | | PMEX (n = 59) | | Control (n = 38) | | P value ($\eta^2$) | |
| --- | --- | --- | --- | --- | --- | --- | --- | --- |
| | Baseline | 16 weeks | Baseline | 16 weeks | Baseline | 16 weeks | Time | Time × Group |
| Weight (kg) | 82.2 ± 11.4 | 81.4 ± 11.1 | 85.7 ± 12.9 | 84.7 ± 12.7 | 87.3 ± 11.1 | 86.9 ± 11.0 | 0.07 (0.02) | 0.31 (0.02) |
| Body mass index (kg m$^{-2}$) | 30.0 ± 2.9 | 29.7 ± 3.0 | 30.5 ± 2.9 | 30.1 ± 2.8 | 31.6 ± 3.0 | 31.4 ± 3.1 | 0.40 (0.01) | 0.50 (0.01) |
| Fat mass (kg) | 27.8 ± 5.4 | 27.0 ± 5.4*† | 28.7 ± 5.8 | 28.0 ± 5.6*† | 29.8 ± 6.4 | 30.4 ± 6.1* | **0.008 (0.05)** | **0.002 (0.09)** |
| Fat-free mass (kg) | 54.4 ± 10.0 | 54.4 ± 9.5 | 57.0 ± 11.4 | 56.4 ± 10.8 | 57.5 ± 10.7 | 56.6 ± 10.0 | **0.037 (0.03)** | 0.13 (0.03) |
| Waist circumference (cm) | 101 ± 10 | 99 ± 10#† | 105 ± 9 | 103 ± 8#† | 107 ± 8 | 108 ± 8 | **0.027 (0.04)** | **<0.001 (0.20)** |
| Glucose (mg dL$^{-1}$) | 109 ± 24 | 108 ± 23 | 113 ± 31 | 114 ± 30 | 112 ± 23 | 111 ± 19 | **<0.001 (0.10)** | 0.28 (0.02) |
| Triglycerides (mg dL$^{-1}$) | 116 ± 52 | 110 ± 43 | 137 ± 77 | 138 ± 70 | 138 ± 69 | 130 ± 64 | **<0.001 (0.14)** | 0.25 (0.02) |
| HDL-cholesterol (mg dL$^{-1}$) | 49 ± 16 | 49 ± 14 | 46 ± 12 | 46 ± 11 | 46 ± 10 | 46 ± 10 | **<0.001 (0.18)** | 0.94 (0.01) |
| Systolic blood pressure (mmHg) | 131 ± 20 | 126 ± 15*† | 127 ± 14 | 126 ± 13 | 127 ± 14 | 130 ± 13 | **<0.001 (0.28))** | **0.019 (0.06)** |
| Diastolic blood pressure (mmHg) | 81 ± 11 | 76 ± 9#† | 80 ± 8 | 77 ± 7#† | 79 ± 8 | 80 ± 7 | **<0.001 (0.28)** | **0.004 (0.08)** |
| MetS Z score | 0.46 ± 0.68 | 0.22 ± 0.58#†‡ | 0.57 ± 0.58 | 0.46 ± 0.51*† | 0.59 ± 0.52 | 0.61 ± 0.52 | 0.72 (0.01) | **<0.001 (0.14)** |
| MetS prescribed medications (n) | 1.19 ± 1.09 | 1.21 ± 1.07 | 1.31 ± 0.79 | 1.31 ± 0.79 | 1.45 ± 1.03 | 1.45 ± 1.03 | 0.09 (0.02) | 0.37 (0.02) |
| Mean arterial pressure (mmHg) | 98 ± 13 | 93 ± 10#† | 96 ± 9 | 93 ± 8* | 95 ± 10 | 96 ± 9 | **<0.001 (0.28)** | **0.004 (0.08)** |
| Insulin ($\mu$U mL$^{-1}$) | 10.5 ± 4.5 | 9.2 ± 3.4*† | 11.5 ± 5.4 | 11.0 ± 4.9 | 12.2 ± 5.1 | 13.0 ± 5.0* | **<0.001 (0.19)** | **0.001 (0.09)** |
| HOMA-IR | 2.9 ± 1.5 | 2.5 ± 1.3*† | 3.2 ± 1.6 | 3.1 ± 1.4 | 3.4 ± 1.6 | 3.5 ± 1.5 | **<0.001 (0.19)** | **0.006 (0.07)** |
| Total cholesterol (mg dL$^{-1}$) | 189 ± 44 | 191 ± 44 | 191 ± 34 | 194 ± 35 | 189 ± 38 | 184 ± 47 | **0.03 (0.03)** | 0.18 (0.03) |
| LDL-cholesterol (mg dL$^{-1}$) | 119 ± 35 | 121 ± 37 | 118 ± 30 | 122 ± 31 | 118 ± 31 | 113 ± 38 | **0.017 (0.04)** | 0.17 (0.03) |
| $\dot{V}_{O_2max}$ (L min$^{-1}$) | 1.93 ± 0.54 | 2.25 ± 0.58#† | 2.11 ± 0.64 | 2.33 ± 0.67#† | 2.09 ± 0.56 | 2.02 ± 0.49 | **<0.001 (0.11)** | **<0.001 (0.25)** |
| $\dot{V}_{O_2max}$ (mL kg$^{-1}$ min$^{-1}$) | 23.3 ± 5.0 | 27.4 ± 6.0#† | 24.4 ± 5.7 | 27.4 ± 6.2#† | 24.2 ± 5.9 | 23.1 ± 4.8* | **<0.001 (0.11)** | **<0.001 (0.30)** |
| Maximal power output (W) | 154 ± 46 | 187 ± 51#† | 173 ± 58 | 199 ± 64#† | 160 ± 44 | 154 ± 41 | **0.002 (0.07)** | **<0.001 (0.40)** |
| Maximal fat oxidation (g min$^{-1}$) | 0.18 ± 0.05 | 0.23 ± 0.08#† | 0.21 ± 0.09 | 0.27 ± 0.11#† | 0.23 ± 0.08 | 0.23 ± 0.09 | **0.002 (0.09)** | **0.01 (0.08)** |

Significant change from baseline within each group (*$P < 0.05$; #$P < 0.001$). Bold significant $p$ values less than 0.05. †Different from Control at the same time point ($P < 0.05$). ‡Different from PMEX at the same time point ($P = 0.021$). Data presented as the mean ± SD.

MetS components were not different among groups. After 16 weeks, there was a significant time effect in the five components of MetS (all $P < 0.05$). However, only a significant time × group interaction effect was found after intervention in waist circumference ($P < 0.001$) and mean arterial pressure ($P = 0.004$). Although, in the Control group, there were no changes in waist circumference or mean arterial pressure, both exercise groups significantly reduced their waist circumference (AMEX: −2.3 cm; 95% CI = −3.30 to −1.65; PMEX: −1.9; 95% CI = −2.57 to −1.19; both $P < 0.001$) and mean arterial pressure (AMEX: −4.9 mmHg; 95% CI = −6.21 to −2.19, $P < 0.001$; PMEX: −2.3; 95% CI = −4.22 to −0.84, $P = 0.004$). After the intervention, the *post hoc* analysis showed waist circumference differences between both exercise groups and Control ($P < 0.001$). However, only the AMEX group lowered mean arterial pressure significantly in comparison to Control ($P = 0.004$). A significant time × group interaction effect was found in MetS Z score ($P < 0.001$). MetS Z score did not change in Control (0.02; 95% CI = −0.05 to 0.12, $P = 0.44$) but was reduced after exercise in both AMEX (−0.24; 95% CI = −0.34 to −0.17, $P < 0.001$) and PMEX (−0.11; 95% CI = −0.17 to −0.04, $P = 0.003$). Improvements in MetS Z score in both exercise groups were larger than in Control (AMEX, $P > 0.001$; PMEX, $P = 0.045$). Furthermore, the reductions in AMEX were also larger than in PMEX ($P = 0.021$) (Fig. 2).

## Additional health parameters

The evolution of systolic and diastolic blood pressure, fasting insulin, HOMA-IR, total cholesterol and LDL-cholesterol are depicted in Table 3. A significant time × group interaction effect was found after the intervention in systolic blood pressure ($P = 0.011$) and diastolic blood pressure ($P = 0.004$), fasting insulin ($P = 0.044$) and HOMA-IR ($P = 0.006$). Although, in PMEX and Control groups, there were no changes after 16 weeks of training in systolic blood pressure and HOMA-IR, the AMEX group lowered their systolic blood pressure (−5.5 mmHg; 95% CI = −7.27 to −1.21, $P = 0.006$) and HOMA-IR (−0.40; 95% CI = −0.19 to −0.78, $P = 0.001$). After 16 weeks of training, fasting insulin was reduced in AMEX (−1.3 $\mu$U mL$^{-1}$; 95% CI = −2.62 to −0.63, $P = 0.002$), whereas it was increased in the Control group (0.8 $\mu$U mL$^{-1}$; 95% CI = 0.06 to 2.15, $P = 0.038$). Therefore, after the intervention, fasting insulin was lower in AMEX than in the Control group ($P = 0.001$), whereas there were no differences between PMEX and Control. After 16 weeks of training, there was no change in diastolic blood pressure in the Control group, whereas both exercise groups improved similarly (AMEX: −4.6 mmHg; 95% CI = −6.08 to −2.40, $P < 0.001$; PMEX: −2.9 mmHg; 95% CI = −4.42 to −1.32, $P < 0.001$). Reductions in diastolic blood pressure in both exercise groups were larger than in Control (both $P < 0.05$). Although there was a significant time effect in total cholesterol ($P = 0.03$) and LDL-cholesterol ($P = 0.017$) concentrations, there was no interaction effect.

## Nutrition and activity levels

Food diaries revealed that pre-exercise meals were similar in macronutrient distribution (47 ± 7%, 34 ± 6% and 18 ± 4% as average for carbohydrate fat and protein). However, the caloric content of the pre-exercise meal in the AMEX group averaged 424 ± 34 kcal as breakfast, which was consumed 45 ± 10 min before the workout, whereas the PMEX group ingested 828 ± 48 kcal as lunch at least 62 ± 15 min before the workout (Table 4). We could not detect significant changes among groups in 24 h calorie intake, macronutrient distribution or physical activity month by month (see Supporting information, Doc. S1) or as reported before or after 16 weeks of training (all $P > 0.05$ for time × group) (Table 4).

## Discussion

In the present study, individuals with MetS were block-randomized and completed a supervised 16 week exercise-training program either in the morning or afternoon (AMEX and PMEX groups) or continued with their habitual physical activity level (Control

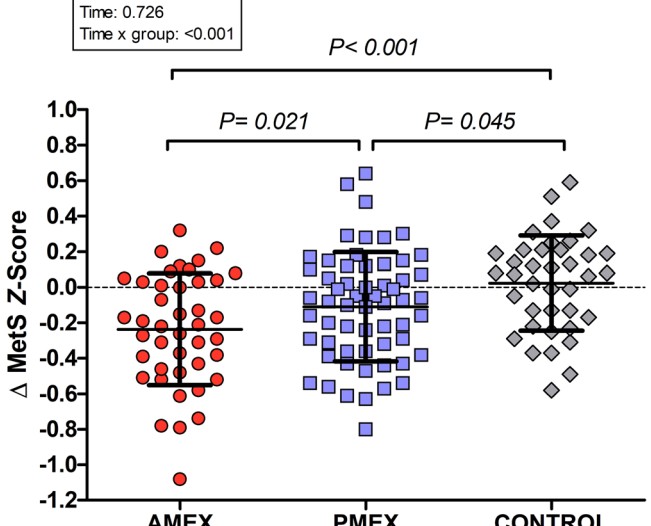

**Figure 2. MetS Z score after intervention**
Changes in MetS Z score after intervention in the AMEX, PMEX or Control group, respectively. Values are presented as dot plots with the mean ± SD. [Colour figure can be viewed at wileyonlinelibrary.com]

**Table 4. Pre-exercise and 24 h caloric intake and physical activity before and after 16 weeks of high-intensity interval training. Data are presented as mean ± SD**

| | AMEX (n = 42) | | PMEX (n = 59) | | Control (n = 38) | | P value ($\eta^2$) | |
| --- | --- | --- | --- | --- | --- | --- | --- | --- |
| | Baseline | 16 weeks | Baseline | 16 weeks | Baseline | 16 weeks | Time | Time × Group |
| Pre-exercise calorie intake (kcal) | 424 ± 34 | 427 ± 31 | 828 ± 48 | 825 ± 40 | – | – | 0.955 (0.01) | 0.591 (0.01) |
| 24 h calorie intake (kcal/day) | 2109 ± 522 | 2090 ± 601 | 2307 ± 445 | 2292 ± 440 | 2207 ± 495 | 2073 ± 530 | 0.104 (0.04) | 0.366 (0.01) |
| % Carbohydrate | 50 ± 10 | 49 ± 8 | 44 ± 7 | 50 ± 8 | 46 ± 5 | 49 ± 7 | 0.101 (0.03) | 0.360 (0.02) |
| % Fat | 35 ± 9 | 34 ± 6 | 33 ± 6 | 29 ± 4 | 35 ± 3 | 33 ± 4 | 0.09 (0.04) | 0.499 (0.01) |
| % Saturated fat | 38 ± 10 | 41 ± 7 | 38 ± 3 | 38 ± 7 | 40 ± 8 | 41 ± 6 | 0.344 (0.01) | 0.642 (0.01) |
| % Protein | 12 ± 4 | 17 ± 3 | 23 ± 5 | 21 ± 3 | 19 ± 2 | 18 ± 3 | 0.244 (0.01) | 0.110 (0.03) |
| Physical activity (steps day$^{-1}$) | 6430 ± 1301 | 6305 ± 1385 | 5898 ± 1275 | 6109 ± 1545 | 5430 ± 313 | 6005 ± 1457 | 0.162 (0.03) | 0.375 (0.01) |
| Time standing (min day$^{-1}$) | 188 ± 119 | 207 ± 95 | 203 ± 113 | 197 ± 100 | 195 ± 88 | 207 ± 101 | 0.248 (0.01) | 0.613 (0.01) |
| Time in supine rest (min day$^{-1}$) | 500 ± 190 | 517 ± 190 | 524 ± 167 | 516 ± 156 | 488 ± 201 | 492 ± 194 | 0.438 (0.01) | 0.656 (0.01) |

group). Their compliance with training was 91% of the supervised sessions and did not differ between EX groups. The 24 h calorie intake and physical activity level were similar among groups and did not change with exercise training in the EX groups. We observed the expected training adaptations in health and fitness parameters when compared to the non-training Control group (Morales-Palomo et al., 2019). Among those, there were small but consistent reductions in waist circumference ($-2.1 \pm 2.8$ cm) and mean arterial pressure ($-3.6 \pm 1.6$ mmHg), as well as increases in CRF (14% in $\dot{V}_{O_2max}$), metabolic fitness (29% in $FO_{max}$) and exercise performance (18% in cycling $W_{max}$). Those improvements were of similar magnitude in the AMEX and PMEX groups (Table 3). However, systolic blood pressure, plasma insulin concentrations and HOMA-IR were improved significantly only in the AMEX group (Table 3). Of note, the MetS $Z$ score, a compound score sensitive to subtle evolution in each MetS component, was improved in AMEX beyond the levels of the PMEX group (Fig. 2). In summary, our data suggest that high-intensity aerobic exercise training for 16 weeks in the morning (AMEX) is more efficient at reducing insulin sensitivity [i.e. insulin concentration and insulin resistance (HOMA-IR)] and systolic blood pressure than similar training in the afternoon (PMEX) in individuals with MetS.

Several studies have explored the influence of the time of day of exercise on weight management, but the findings are mixed. We found that both training groups experienced a similar reduction in body fat (AMEX, $-0.8 \pm 2.1$ kg; PMEX, $-0.6 \pm 2.2$ kg; $P < 0.05$) and waist circumference (AMEX, $-2.3 \pm 3.2$ cm; PMEX, $-1.9 \pm 2.3$ cm; $P < 0.05$). Our results are in agreement with other studies that reported no differences in weight loss and body composition between morning and evening exercise groups (Blankenship et al., 2021; Janssen et al., 2022). By contrast, two randomized controlled trials and one review reported larger reductions in body weight in participants who performed exercise in the morning (Alizadeh et al., 2017; Schumacher et al., 2020; Willis et al., 2020). Conversely, two non-randomized trials reported that morning exercise resulted in a smaller loss of body fat than when exercising in the afternoon or evening (Di Blasio et al., 2010; Mancilla et al., 2021).

Both exercising groups maintained caloric intake and physical activity unchanged during the 16 week intervention (Table 4). On average, participants incurred $388 \pm 83$ kcal of energy deficit per exercise session (Table 2), which likely induced the 0.8–0.6 kg of body fat losses reported. An energy deficit of 7700 kcal is required to lose 1 kg of body weight (Hall et al., 2012). Based on these data, an average weight loss of more than 2.5 kg was expected after the 16 week training intervention in the AMEX and PMEX groups. However, we observed

only modest changes in the exercise groups (AMEX, −0.8; PMEX,−1.0 kg). The difference between expected and actual weight loss is well-documented in the literature and may be related to other physiological or behavioural compensatory adaptive responses to oppose the energy deficit (Melanson et al., 2013). Nevertheless, in the absence of dietary caloric restriction, the effect of exercise timing (16 weeks) does not reflect in different reductions in body weight or body composition which suggests that the compensatory effects on appetite and energy intake were not different between exercising groups (AMEX *vs.* PMEX).

It has been speculated that the beneficial effects of exercise on glycaemic control and insulin sensitivity could be partly a result of the resetting of circadian rhythms (Gabriel & Zierath, 2019). In response to the 16 weeks of intense aerobic exercise, there were no statistically significant improvements in fasting blood glucose in any of the exercising groups (AMEX or PMEX; time × group effect, $P = 0.28$) (Table 3). This finding suggests that, in middle-aged MetS individuals, 16 weeks of exercise, without the addition of a hypocaloric diet or glucose-lowering medication, is not sufficient to improve glycaemic control (i.e. fasting glucose). Interestingly, exercising in the morning (AMEX group) lowered fasting insulin levels by 12% ($-1.3\ \mu$IU·ml$^{-1}$; time × group effect, $P = 0.001$), whereas insulin concentrations increased by 7% over baseline values in the Control group ($0.8\ \mu$IU mL$^{-1}$). As a result, HOMA-IR was reduced by 14% in the AMEX group ($-0.4$; time × group effect, $P = 0.006$), whereas it did not improve in the PMEX group. Although HOMA-IR is not a clinical index typically used to monitor glycaemic control, it provides important information on the progression toward type 2 diabetes. Insulin resistance precedes, often by many years, the development of type 2 diabetes (Pradhan et al., 2003; Saad et al., 1989) and primary prevention of type 2 diabetes necessitates early detection of insulin resistance.

Our findings coincide partially with those of a study by Teo et al. (2020) who enrolled participants overweight with and without type 2 diabetes and reported overall improvements in glycaemic response regardless of the timing of 12 weeks of multimodal exercise. However, they reported a trend toward the superiority of morning training only among participants with type 2 diabetes. Conversely, other studies have shown that, in men with metabolic disarrangements (Mancilla et al., 2021) or type 2 diabetes (Savikj et al., 2019), high-intensity exercise in the afternoon is more efficacious in improving blood glucose than a high-intensity exercise in the morning. Of note, one of those studies measured the effects of only 2 weeks of high-intensity interval training (Savikj et al., 2019) and the other performed a non-randomized concurrent training intervention (Mancilla et al., 2021). Two recent meta-analyses have shown that the timing of exercise does not affect glycaemic control improvements to a bout of exercise (Sevilla-Lorente et al., 2023) or after a full exercise program (i.e. >2 weeks) (Galan-Lopez & Casuso, 2023).

It is noteworthy that the timing of exercise concerning meals could affect glycaemic control (Martínez-Montoro et al., 2023). Postprandial hyperinsulinaemia and associated peripheral insulin resistance are key drivers of type 2 diabetes and associated cardiovascular diseases (Reaven, 1988). In overweight and obese men, only exercise training before carbohydrate ingestion reduced postprandial insulinaemia and increased the oral glucose tolerance test-derived estimate of peripheral insulin sensitivity (Edinburgh et al., 2019). Although, in our experimental design, all the participants performed the exercise ∼1 h after meal ingestion (i.e. breakfast or lunch), the pre-exercise meal in the afternoon group contained twice as many kilocalories as the meal of the morning group (PMEX, 424 kcal *vs.* AMEX, 828 kcal) (Table 4). Therefore, the improvements in insulin sensitivity in the AMEX group may be a result of the promotion of lipid utilization and reduced postprandial insulinaemia.

Resting blood pressure presents a circadian pattern with a morning increase, a marked postprandial valley and a deeper descent during night sleeping (Biaggioni, 2008; Ramirez-Jimenez et al., 2022). Blood pressure is regulated mainly by the neurohumoral system, which includes the renin–angiotensin–aldosterone system, the sympathetic nervous system and the immune system. These regulatory mechanisms allow the human body to respond to exercise with rapid postexercise hypotension (Halliwill, 2001). Supervised aerobic exercise training results in reductions in resting blood pressure according to the magnitude of hypertension of the individual undergoing training according to Wilder's principle (Mora-Rodriguez et al., 2022). A blood pressure lowering effect occurs in response to an acute bout of exercise (i.e. post-exercise hypotension) (Morales-Palomo et al., 2017), in response to a full training program (i.e. 16 weeks) (Morales-Palomo et al., 2019), and remains reduced when several training programs are repeated over the years (Morales-Palomo et al., 2017). However, it is not clear whether the time of day at which individuals exercise-train could affect the magnitude of the blood pressure-lowering effect of exercise.

We found that, after 16 weeks of a high-intensity aerobic exercise program, both exercise groups improved diastolic blood pressure regardless of the time-of-day training (Table 3). Of note, after conducting *post hoc* analyses, the lowering effects of 16 weeks of exercise training on mean arterial pressure and systolic blood pressure were only statistically significant in the AMEX *vs.* Control group ($P = 0.004$ and $P = 0.015$, respectively). Our results are partially opposite with a randomized controlled trial in men with hypertension, showing that only afternoon training produced clinic and ambulatory hypotensive effects (Brito et al., 2019). However, our data coincide with another randomized crossover study

in prehypertensive men where the post-exercise hypotensive effect was greater after exercise in the morning than in the evening (de Brito et al., 2015). Although it is unclear what factors contribute to AMEX's larger blood pressure lowering effect, it could be that exercise after the blood pressure night dip might attenuate the circadian-induced morning rise in blood pressure. We have recently reported that intense aerobic exercise improves nocturnal blood pressure dipping in medicated hypertensive patients (Ramirez-Jimenez et al., 2022). Therefore, in the current cohort that included 58% of hypertensive subjects, morning exercise training has a larger effect on reducing systolic blood pressure after training maybe by extending nocturnal blood pressure dipping.

Traditionally MetS diagnosis is calculated based on exceeding a threshold for each one of the 5 MetS components (i.e. hypertension, hyperglycaemia, abdominal obesity and dyslipidaemia). However, this threshold approach is not sufficiently sensitive to detect subtle progression in each of the factors (Gurka et al., 2014). Thus, we calculated a compound MetS $Z$ score to assess the continuous rather than dichotomous (pass/not pass) evolution of each MetS component. Z score was calculated as the difference between the subject's value and threshold value divided by the group SD for each MetS criterion. MetS $Z$ score summarizes in one number the extent of the deviation from a healthy value of the five MetS components together. The positive short-term effects of an exercise-training program on MetS $Z$-score had been previously reported (Bateman et al., 2011; Mora-Rodriguez et al., 2016). However, to our knowledge, this is the first study to seek the effect of time of day of high-intensity aerobic exercise training on MetS $Z$-score. In our data, pivotal to a larger reduction in MetS $Z$-score in AMEX *vs*. PMEX was the drop in systolic blood pressure followed by subtle changes in blood glucose and triglyceride concentration. Those changes, not statistically significant, contributed to the differences in the $Z$ score (AMEX, −0.24 *vs*. PMEX, −0.11; $P = 0.021$) (Fig. 2).

It is unclear whether the training time of day could influence cardiorespiratory and metabolic training adaptations. Comparisons of $\dot{V}_{O_2max}$ and fat oxidation have also been made between AM and PM, but the results were inconsistent, with some studies showing higher improvements in the PM training group (Kang et al., 2023), whereas others revealed no time effect (Brooker et al., 2023). Detecting time of day variations in endurance performance is more challenging because the results can be influenced by physiological adjustments that may not always follow circadian rhythms. The initial CRF level of our MetS participants was in the lower 15 percentile for their age and sex ($\dot{V}_{O_2max}$ of 24 mL $O_2$ kg$^{-1}$ min$^{-1}$) (Table 3) (Magal et al., 2017). It appears that, for these initially deconditioned individuals, a 16 week exercise program strongly improves CRF and fat metabolism shadowing any effect of training time of day. On the other hand, $FO_{max}$ was similarly improved in AMEX and PMEX without an effect on the training time of day (Table 3). This suggests that this metabolic adaptation to training (increase peak fat oxidation) is not altered by training time of day and rather responds to frequency, intensity, duration and type of exercise.

The present study has strengths and limitations. The strengths are that, in addition to the randomized-controlled design, the sample size was larger than in other similar studies, boosting statistical power and the trustworthiness of the results. Compliance with exercise intensity and duration of each session was directly supervised. Physical activity outside the training intervention and 24 h caloric intake were monitored to assess if compensatory behaviour took place between groups. Conversely, the study is not free of limitations. We did not assess participants' circadian preference for the timing of various activities (i.e. morningness *vs*. eveningness questionnaire). We hope that subject randomization resulted in an even distribution of those individual types in the three experimental groups. Our participants were middle-aged MetS patients with sufficient motivation to complete 16 weeks of training. Similar results may not be obtained in a no-supervised group or less motivated patients. The 24 h ambulatory blood pressure provides a more comprehensive assessment of blood pressure than clinical blood pressure (Staplin et al., 2023). Thus, the measure of ambulatory blood pressure could have provided us with more information about the circadian blood pressure pattern. Another limitation was that the intervention only included aerobic exercise training, whereas evidence is growing that the inclusion of strength training could further improve MetS.

In summary, it has been hypothesized that MetS emerges because of a derangement of the circadian rhythm. On the other hand, exercise could be a solid cue for realigning those rhythms. In a randomized-controlled fashion, we manipulated exercise training time of day in individuals with MetS and measured cardiovascular and metabolic responses to training. We corroborated that 16 week of supervised high-intensity interval aerobic exercise (AMEX and PMEX), without dietary restriction, improved cardiorespiratory and metabolic fitness and body composition compared to a non-exercise Control group. The findings support the concept that exercise training, regardless of the time of day, improves the fitness and health of individuals with MetS. The novel finding was that systolic blood pressure, blood insulin concentration and insulin resistance were improved further in the morning than in the afternoon training group (AMEX *vs*. PMEX). Our data suggest that morning training is a somewhat more effective strategy to maximize health promotion in individuals with MetS.

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

## Additional information

### Data availability statement

The data that support the findings of this study are available from the corresponding author upon reasonable request.

## Competing interests

The authors declare that they have no competing interests.

## Author contributions

F.M.P, R.M.R., A.M.C. and L.A.J. were responsible for the conceptualization and design of the study. F.M.P., J.F.O., A.M.C., D.M.G. and L.A.J. were responsible for the acquisition of data. F.M.P., R.M.R., AMC, J.F.O. and L.A.J. were responsible for the analysis and interpretation of data. F.M.P. and R.M.R. were responsible for writing the original draft. F.M.P., R.M.R., A.M.C., D.M.G. and L.A.J. were responsible for reviewing and editing. F.M.P., J.F.O. and R.M.R. were responsible for supervision. All authors have approved the final version of the manuscript and agree to be accountable for all aspects of the work. All persons designated as authors qualify for authorship, and all those who qualify for authorship are listed.

## Funding

Spanish Ministry of Science and Innovation (PID2020-116159RB- IOO MCIN/AEI/10.13039/501 100 011 033). The granting agency has no role in the design, execution or reporting of the results of this study.

## Keywords

circadian rhythms, exercise timing, metabolic syndrome

## Supporting information

Additional supporting information can be found online in the Supporting Information section at the end of the HTML view of the article. Supporting information files available:

**Peer Review History**
**Supporting information**

