## [Peer Review History · The Journal of Physiology]

Efficacy of morning versus afternoon aerobic exercise training on reducing metabolic syndrome components: A randomized controlled trial

Felix Morales-Palomo, Alfonso Moreno-Cabañas, Laura Alvarez-Jimenez, Diego Mora-Gonzalez, Juan Fernando Ortega, and Ricardo Mora-Rodriguez

DOI: 10.1113/JP285366

Corresponding author(s): Ricardo Mora-Rodriguez (ricardo.mora@uclm.es)

Review Timeline:

Submission Date:	25-Jul-2023
Editorial Decision:	08-Aug-2023
Revision Received:	28-Aug-2023
Editorial Decision:	22-Sep-2023
Revision Received:	03-Oct-2023
Accepted:	16-Oct-2023

Senior Editor: Paul Greenhaff

Reviewing Editor: Bettina Mittendorfer

Transaction Report:

Dear Dr Mora-Rodriguez,

Re: JP-RP-2023-285366 "Efficacy of morning versus evening exercise training on reducing metabolic syndrome components: A randomized controlled trial" by Felix Morales-Palomo, Alfonso Moreno-Cabañas, Laura Alvarez-Jimenez, Diego Mora-Gonzalez, Juan Fernando Ortega, and Ricardo Mora-Rodriguez

Thank you for submitting your manuscript to The Journal of Physiology. It has been assessed by a Reviewing Editor and by 2 expert referees and we are pleased to tell you that it is potentially acceptable for publication following satisfactory major revision.

LANGUAGE EDITING AND SUPPORT FOR PUBLICATION: If you would like help with English language editing, or other article preparation support, Wiley Editing Services offers expert help, including English Language Editing, as well as translation, manuscript formatting, and figure formatting at www.wileyauthors.com/eoo/preparation. You can also find resources for Preparing Your Article for general guidance about writing and preparing your manuscript at www.wileyauthors.com/eoo/prepresources.

REVISION CHECKLIST:

Please upload two versions of your manuscript text: one with all relevant changes highlighted and one clean version with no changes tracked. The manuscript file should include all tables and figure legends, but each figure/graph should be uploaded as separate, high-resolution files. The journal is now integrated with Wiley's Image Checking service. For further details, see: <https://www.wiley.com/en-us/network/publishing/research-publishing/trending-stories/upholding-image-integrity-wileys->

image-screening-service

We look forward to receiving your revised submission.

Yours sincerely,

Paul Greenhaff
Senior Editor
The Journal of Physiology

REQUIRED ITEMS

- Author photo and profile. First (or joint first) authors are asked to provide a short biography (no more than 100 words for one author or 150 words in total for joint first authors) and a portrait photograph. These should be uploaded and clearly labelled with the revised version of the manuscript. See Information for Authors for further details.
- Your manuscript must include a complete Additional Information section.
- The Journal of Physiology funds authors of provisionally accepted papers to use the premium BioRender site to create high resolution schematic figures. Follow this link and enter your details and the manuscript number to create and download figures. Upload these as the figure files for your revised submission. If you choose not to take up this offer we require figures to be of similar quality and resolution. If you are opting out of this service to authors, state this in the Comments section on the Detailed Information page of the submission form. The link provided should only be used for the purposes of this submission. Authors will be charged for figures created on this premium BioRender account if they are not related to this manuscript submission.
- Please upload separate high-quality figure files via the submission form.
- Please ensure that any tables are in Word format and are, wherever possible, embedded in the article file itself.
- Papers must comply with the Statistics Policy: https://jp.msubmit.net/cgi-bin/main.plex?form_type=display_requirements#statistics.

In summary:

- If n {less than or equal to} 30, all data points must be plotted in the figure in a way that reveals their range and distribution. A bar graph with data points overlaid, a box and whisker plot or a violin plot (preferably with data points included) are acceptable formats.
- If $n > 30$, then the entire raw dataset must be made available either as supporting information, or hosted on a not-for-profit repository e.g. FigShare, with access details provided in the manuscript.
- 'n' clearly defined (e.g. x cells from y slices in z animals) in the Methods. Authors should be mindful of pseudoreplication.
- All relevant 'n' values must be clearly stated in the main text, figures and tables.
- The most appropriate summary statistic (e.g. mean or median and standard deviation) must be used. Standard Error of the Mean (SEM) alone is not permitted, unless justified and presented alongside confidence intervals.
- Exact p values must be stated. Authors must not use 'greater than' or 'less than'. Exact p values must be stated to three significant figures even when 'no statistical significance' is claimed.
- Please include an Abstract Figure file, as well as the figure legend text within the main article file. The Abstract Figure is a piece of artwork designed to give readers an immediate understanding of the research and should summarise the main conclusions. If possible, the image should be easily 'readable' from left to right or top to bottom. It should show the physiological relevance of the manuscript so readers can assess the importance and content of its findings. Abstract Figures should not merely recapitulate other figures in the manuscript. Please try to keep the diagram as simple as possible and

without superfluous information that may distract from the main conclusion(s). Abstract Figures must be provided by authors no later than the revised manuscript stage and should be uploaded as a separate file during online submission labelled as File Type 'Abstract Figure'. Please ensure that you include the figure legend in the main article file. All Abstract Figures should be created using BioRender. Authors should use The Journal's premium BioRender account to export high-resolution images. Details on how to use and access the premium account are included as part of this email.

EDITOR COMMENTS

Reviewing Editor:

Both reviewers found merit in the paper but noticed some issues that need to be addressed

Senior Editor:

This manuscript has been considered by two expert reviewers and a reviewing editor with expertise in the field. All are in agreement that there are merits in the research. However, Reviewer 2 believes there are several limitations with the manuscript that currently limit the scientific clarity and impact of the work described, which the reviewing editor and senior editor agree are valid. Of note, amongst a number of important points, the clinical trial registry that the authors refer in the paper appears not to be the study that is described in the manuscript and the statistical analysis conducted needs to be very much re-evaluated. The manuscript therefore requires major revision if it is to be considered further. Hopefully the authors feel they can undertake the necessary major revisions.

REFEREE COMMENTS

Referee #1:

This is an interesting, overall clinically relevant paper. I also think the Ms is overall well written and the stats seem to be OK. So congrats to the authors in general. The results are largely novel.

My comments are minor in essence and rather 'addressable'.

1. Intro is a bit too long in my opinion. I could be maybe shortened by 25-30% and made more 'focused'.
2. The authors wrote that the 'Participants were previously sedentary (<150 min-wk-1 of the moderate-intensity activity assessed by 7-d IPAQ (Craig et al., 2003))'. I think 'inactive' is the correct term here instead of 'sedentary'.
3. I am not a fan of using so many nonstandard abbreviations (e.g., CRF is fine but I would spell in full most of the proposed abbreviations). To me, the 'philosophy' of abbreviations is to ease readability and/or to save words. None of this apply for instance to the use of 'CONT' instead of control. I suggest changing 'AMEX' and all this to simply 'morning (or evening) exercise' or 'a.m. (or p.m.) exercise'. Since this is largely a clinical paper, I would use 'clinical writing', where abbreviations are kept to a strict minimum, not to mention in the Introduction section (where, if possible, I would simply refrain from using abbreviations).
4. Tables: To make tables more attractive (and even more scientifically correct), I would not use decimals for those variables where decimals are beyond the accuracy of the assessment itself (e.g., no decimals should be used at all for SBP/DBP, cholesterol etc.). I would also avoid decimals for variables with 3 digits (e.g., maximal power output data are perfect). Anyway, could the authors replace some tables by figures? (optional).
5. Is there a possibility of conducting subanalyses by sex? The N is large enough to do so. If the results don't change essentially, just add this.
6. Limitations: It should be noted that the authors did not measure ambulatory (24 h) blood pressure. This is certainly the variable I am missing the most since 24h blood pressure (especially night and systolic night blood pressure) is by far much more informative than 'clinic' or 'office' measurements (e.g., see recent paper by authors' compatriots in Lancet (Ruilope's group)).
7. Limitations: to some extent, same for lack of DXA-determined body composition (it would have been great to have proxies for visceral adipose tissue).

Referee #2:

This study investigates the impact of timing of a 16-week HIIT intervention on cardiovascular risk factor in people with MetS.

The research query is intriguing and holds significant promise. Nonetheless, the study is accompanied by several limitations that impede the applicability and reliability of its findings.

Authors refer to evening exercise in several places in the manuscript, but exercise sessions were between 16:00 to 18:00, and this is not evening, at least in Spain. It should be called afternoon. please clarify this aspect all through the manuscript.

Methods

Authors mention "Body weight stability in the four months before study enrolment was an additional requirement" - please indicate what do you mean by 'body weight stability'.

Authors mention that "This study is the main part of an ongoing clinical trial (NCT04477590)", but the study that is registered refers to a "Cross-over randomized double-blinded, pretest-posttest control group experimental design. The project will be developed in a single center with the collaboration of the regional public health system (SECAM)". The present study has a totally different study design. Are these two studies the same? Actually, in the registry indicates "Specifically, we will study if the cardiovascular and metabolic adaptations to aerobic training that result in amelioration of metabolic syndrome factors are potentiated by correct timing of meal and medicine around exercise training time". Details on arms and interventions are not the same. This study aim does not match with the aim of the current study and this should be corrected and specified. Please clarify.

The randomisation was unbalanced: 1 (AMEX): 1.5(PMEX): 1(CONT) ratio. Can the authors provide more details and rationale for this?

Diet: It was monitored during the intervention (monthly). Can the authors provide if there were any changes during the intervention - month by month? Similarly, physical activity was measured for 48h every month. Can the authors provide this data (monthly) to better understand the impact of the intervention on these lifestyle habits?

Participants were required to consume a meal at least 1 h before the start of each training session. Was this confirmed by any way? is it likely that AMEX exercised in fasting conditions?

Exercise training: 4 x 4-minute intervals at 90% of HRMAX interspersed with a 3-minute active recovery at 70% of HRMAX. Was there any training adaptation? this exercise load seem too hard as a starting point for a sedentary person. Please, describe.

Please describe how was HRMAX reevaluated.

Blood sampling. When was blood samples taken? how many days after the last exercise session? please specify.

The equation to determine MetS Z-Score included MAP instead of the single values of SPB or DPB as indicated in the definitions. Can the authors specify why? please, provide a rationale on this.

Statistical analysis

There is need to include a detailed description of the main outcome variable and statistical power calculations.

The estimations of the effect size are not clear. Please, indicate the main outcome variable, the expected changed and the comparisons you intend to conduct.

The authors should control for multiple comparisons. As presented, the authors make comparisons between control vs exercise (including both exercise groups with no rationale to do so), control vs morning, control vs evening, evening vs morning. Please, indicate whether there is enough power to do so.

The sample combines both men and women, but they may respond differently to the exercise intervention.

It would be appropriate to conduct linear mixed models.

Include details on how was sex treated.

Please, describe the statistical models more in detailed. Dependent variables, etc. Was the changed included in the model? the post-intervention value?

Covariates. only the baseline values was included? but if baseline levels were similar across groups, is this correct to do it?

Information about how were missing values treated should be included.

Results

The authors refer to another study to indicate that women responses did not differ from men's.

responses? what about in the current study? the authors should provide data by sex in the current study at least in supplementary file and conduct formal sex-interaction analyses.

Table 1 and 2. The P value is from the t test or from the chi-square, right? I believe the authors are not providing the chi-square but the P value from the chi-square test. Please, indicate more clearly. Also include its description in the statistical analysis. Is randomisation is correct, to compared baseline values among groups is not needed nor correct

Can the authors specify if the Avg heart rate is from the session?

Can the authors provide the Avg heart rate of the HIIT intervals? was the 90% achieved always?

Was VO2 measured in every exercise session? please specify and describe in methods

Please, describe in detail how was energy expenditure determined.

Discussion

The summary of the first paragraph of the discussion is not fully correct, as no all CVD risk markers differ between AMEX and PMEX.

Please, discuss about the clinical meaning of the differences observed between AMEX and PMEX. Do the authors genuinely hold the belief that the observed "minor" changes hold clinical significance substantial enough to warrant recommending morning exercise for individuals with MetS?

END OF COMMENTS

Confidential Review

25-Jul-2023

RESPONSES TO REVIEWER #1

We appreciate the time devoted by the reviewer to comment on the manuscript. After incorporation of all the corrections we believe that the manuscript is improved. Please find below an itemized response to your comments. Changes in the manuscript are highlighted in red.

Referee #1:

This is an interesting, overall clinically relevant paper. I also think the Ms is overall well written and the stats seem to be OK. So congrats to the authors in general. The results are largely novel.

My comments are minor in essence and rather 'addressable'.

1. Intro is a bit too long in my opinion. I could be maybe shortened by 25-30% and made more 'focused'.

RESPONSE: We agree, now the introduction is more focused and reduced by 182 words (-24%).

2. The authors wrote that the 'Participants were previously sedentary (<150 min·wk-1 of the moderate-intensity activity assessed by 7-d IPAQ (Craig et al., 2003))'. I think 'inactive' is the correct term here instead of 'sedentary'

RESPONSE: You are correct that as reported in Table 4, participants averaged 5919 steps per day and therefore should be considered inactive instead of 'sedentary'.

Now in the METHODS section, we have changed sedentary to inactive: "Participants were previously **inactive** (<150 min·wk-1 of the moderate-intensity activity assessed by 7-d IPAQ (Craig et al., 2003)).

3. I am not a fan of using so many nonstandard abbreviations (e.g., CRF is fine but I would spell in full most of the proposed abbreviations). To me, the 'philosophy' of abbreviations is to ease readability and/or to save words. None of this apply for instance to the use of 'CONT' instead of control. I suggest changing 'AMEX' and all this to simply 'morning (or evening) exercise' or 'a.m. (or p.m.) exercise'. Since this is largely a clinical paper, I would use 'clinical writing', where abbreviations are kept to a strict minimum, not to mention in the Introduction section (where, if possible, I would simply refrain from using abbreviations).

RESPONSE: Now highlighted in red, throughout the manuscript, tables, and figures we have reduced the use of abbreviations (i.e., blood pressure (BP), systolic blood pressure (SBP), diastolic blood pressure (DBP), mean arterial pressure (MAP), heart rate (HR), type 2 diabetes (DMT2)).

We have also changed CONT to CONTROL throughout. However, the AMEX – PMEX is common use in the "time-of-day" scientific literature (PMID: 31479004, PMID: 36502286) and we would like to keep it as it is.

4. Tables: To make tables more attractive (and even more scientifically correct), I would not use decimals for those variables where decimals are beyond the accuracy of the assessment itself (e.g., no decimals should be used at all for SBP/DBP, cholesterol etc.). I would also avoid decimals for variables with 3 digits (e.g., maximal power output data are perfect).

Anyway, could the authors replace some tables by figures? (optional).

RESPONSE: Thanks for the comment. Now Table 4 has been modified by reducing decimals.

5. Is there a possibility of conducting subanalyses by sex? The N is large enough to do so. If the results don't change essentially, just add this.

RESPONSE: The percentage of women participating in each group was similar (AMEX 38%; PMEX 36% and CONTROL 32%; $p=0.17$) and all women were postmenopausal without hormonal treatment. Thus, it is unlikely that a sex factor could be altering the reported a.m. vs. p.m. responses. Further, we have recently reported (PMID: 31821339) that the metabolic and cardiovascular responses to 4 months of HIIT in MetS patients (without controlling for "time-of-day") are similar in men and women.

Below is a statistical analysis by sex of the main responses from the current study.

	Women (n=49)		Men (n=90)		p-value	
	Pre	Post	Pre	Post	Time	Time x Sex
Body weight (Kg)	75.6 ± 8.4	75.3 ± 8.6	90.3 ± 10.5	89.2 ± 10.4	<0.001	0.061
MetS Z-score	0.57 ± 0.68	0.50 ± 0.66	0.53 ± 0.55	0.39 ± 0.49	<0.001	0.201
VO _{2MAX} (L·min ⁻¹)	1.49 ± 0.27	1.66 ± 0.30	2.35 ± 0.48	2.53 ± 0.57	<0.001	0.696
VO _{2MAX} (mL·kg ⁻¹ ·min ⁻¹)	19.9 ± 3.6	22.2 ± 4.4	26.3 ± 5.1	28.5 ± 5.7	<0.001	0.679

Now in the RESULTS section, we have also included the p-values of the interaction time by sex of the main variables of the study: "Women participants comprised 35% of all subjects. Data were analyzed without sex differentiation since all women were postmenopausal and were not taking hormonal replacement (Guio de Prada et al., 2019), and their responses did not differ from men's responses in primary and secondary outcomes of the study (time x sex interaction; MetS Z score (SD), $p=0.20$; body weight (Kg), $p=0.06$; VO_{2MAX} (L·min⁻¹), $p=0.70$).

6. Limitations: It should be noted that the authors did not measure ambulatory (24 h) blood pressure. This is certainly the variable I am missing the most since 24h blood pressure (especially night and systolic night blood pressure) is by far much more informative than 'clinic' or 'office' measurements (e.g., see recent paper by authors' compatriots in Lancet (Ruilope's group)).

RESPONSE: We agree that ambulatory blood pressure provides a more comprehensive assessment of blood pressure and better predicts adverse health outcomes. Unfortunately, ambulatory blood pressure was not measured in this study. However, our clinical blood pressure was measured under the most rigorous conditions as reflected in the Methods: "After 10 minutes of supine rest, blood pressure was measured in triplicate using a calibrated ECG-gated electro-sphygmomanometer (Tango, Suntec Medical; NC)".

However, we concur with the reviewer, and now in the limitation paragraph, we have included:

“Conversely, the study is not free of limitations. Our participants were middle-aged MetS patients with enough motivation to complete 16 weeks of training. Similar results may not be obtained in a no-supervised group or less motivated patients. **24-hour ambulatory blood pressure provides a more comprehensive assessment of blood pressure than clinical blood pressure (Staplin *et al.*, 2023). Thus, the measure of ambulatory blood pressure could have provided us with more information about the circadian blood pressure pattern.** Another limitation was that the intervention only included aerobic exercise training, while evidence is growing that the inclusion of strength training could further improve MetS.

7. Limitations: to some extent, same for lack of DXA-determined body composition (it would have been great to have proxies for visceral adipose tissue).

RESPONSE: We agree with the reviewer that analyzing body composition with DXA would have given us valuable information on the effect of exercise timing on visceral adipose tissue. Perhaps that could have unveiled a link to the improved insulin. We discussed the possibility of using a DXA from colleagues on campus. However, the apparatus lacked the specific software that allows to dissection of visceral adipose tissue from abdominal fat, and the investment to get one was over 7000 euros. Nevertheless, the loss of fat mass measured by bioelectrical impedance analysis was small and similar in both groups (0.7 and 0.8 kg). In our humble opinion, it is unlikely that differences in visceral fat mass loss could have emerged (statistical differences) even if we had used DXA technology.

Referee #2:

RESPONSES TO REVIEWER #2

We truly appreciate the time the reviewer devoted on correcting our manuscript. We have incorporated most of your suggestions into this new version of the manuscript, which, in our opinion, is improved. Please find below an itemized response to your comments. Changes in the manuscript are highlighted in red.

This study investigates the impact of timing of a 16-week HIIT intervention on cardiovascular risk factor in people with MetS.

The research query is intriguing and holds significant promise. Nonetheless, the study is accompanied by several limitations that impede the applicability and reliability of its findings.

Authors refer to evening exercise in several places in the manuscript, but exercise sessions were between 16:00 to 18:00, and this is not evening, at least in Spain. It should be called afternoon. please clarify this aspect all through the manuscript.

RESPONSE: We agree with the reviewer, we have changed the evening to afternoon throughout the manuscript (highlighted in red).

Methods

Authors mention "Body weight stability in the four months before study enrolment was an additional requirement" - please indicate what do you mean by 'body weight stability'.

RESPONSE: We agree with the reviewer that body weight stability should be better described. Now in the METHODS section: "Body weight stability in the four months before study enrollment was an additional requirement (gain or loss <4 kg)".

Authors mention that "This study is the main part of an ongoing clinical trial (NCT04477590)", but the study that is registered refers to a "Cross-over randomized double-blinded, pretest-posttest control group experimental design. The project will be developed in a single center with the collaboration of the regional public health system (SECAM)". The present study has a totally different study design. Are these two studies the same? Actually, in the registry indicates "Specifically, we will study if the cardiovascular and metabolic adaptations to aerobic training that result in amelioration of metabolic syndrome factors are potentiated by correct timing of meal and medicine around exercise training time". Details on arms and interventions are not the same. This study aim does not match with the aim of the current study and this should be corrected and specified. Please clarify.

RESPONSE: While it is true that clinical trial NCT04477590 does not fully match the study design, the outcome measures, and the allocation (randomized), treatment (exercise training using HIIT), and population (metabolic syndrome patients according to the IDF, 2009 consensus) in the clinical trial, describes this study. After posting clinical trial NCT04477590, we realized that before manipulation of meal and medicine timing, we should address if exercise timing itself, would affect the main outcomes. At the time of posting the study (07-15-2020) the literature on training effects of "time-of-day" was emerging and neither we nor the

reviewers of the grant considered that factor. As soon as COVID-19 allowed us to start recruiting participants (September 2020) we started to random-allocation between a.m. and p.m. groups. Since we were still studying timing, and the post-pandemic situation did not guarantee that the study was going to be completed, we did not submit a new version of clinical trials.

We have now updated the clinical trial registry to better reflect and include the data of this paper (below in bold).

Objective: The purpose is to study in a group of adults with metabolic syndrome and obesity, the effects of **altering timing between exercise training**, meals, and their habitual medication.....

Brief Summary: To analyze the effects of altering the time of ingestion of participants' habitual medication (i.e., metformin, statins, ARAI/IACE) and meals around the time of exercise training (exercise fasted or fed) on the improvement of metabolic syndrome factors (hypertension, insulin sensitivity, dyslipidemia, and obesity). **There will be a preliminary study of the effects of training "time-of-day" on the primary study outcomes.**

Methods and design: Cross-over randomized double-blinded, pretest-posttest control group experimental design. The project will be developed in a single center with the collaboration of the regional public health system (SECAM). **There will be a preliminary study of the effects of training "time-of-day" on three parallel groups of individuals.**

Now in the methods section, it is described as: **"This is a substudy part of a larger clinical trial evaluating the effects of interactions of medicine and exercise with meal timing in individuals with Mets (ClinicalTrials.gov Identifier: NCT04477590)".**

The randomization was unbalanced: 1 (AMEX): 1.5(PMEX): 1(CONT) ratio. Can the authors provide more details and rationale for this?

RESPONSE: The reason why we allowed the PMEX group to outnumber the other groups is to allow a better representation of the exercise trend in our region of the world (South Europe; Survey of sporting habits in Spain 2020 (<https://www.culturaydeporte.gob.es/servicios-al-ciudadano/estadisticas/deportes/anuario-de-estadisticas-deportivas.html>)).

Now in the METHODS section, the randomization is described in more detail: "Following baseline testing, participants were randomized to one of three groups; morning exercise (AMEX), **afternoon** exercise (PMEX), or waitlist control (**CONTROL**) at a 1 (AMEX): 1.5 (PMEX): 1 (**CONTROL**) ratio using a block randomized (by sex, age, and BMI) controlled design. **We increased participants in the PMEX group to represent that more individuals in our country exercise after the working day shift (typically between 07:00–15:00 h.) and the ingestion of the main meal.**

Diet: It was monitored during the intervention (monthly). Can the authors provide if there were any changes during the intervention - month by month? Similarly, physical activity was measured for 48h every month. Can the authors provide this data (monthly) to better understand the impact of the intervention on these lifestyle habits?

RESPONSE: You are right that the diet and physical activity data presented in Table 4, only show the values at baseline and 16 weeks. There were no significant changes monthly (see table below) in caloric intake or physical activity:

	AMEX (n=42)				p-value
	Baseline	4 weeks	8 weeks	16 weeks	
Pre-exercise calorie intake (kcal)	424±34	424±61	419±19	427±31	0.34
24-h calorie intake (kcal/day)	2109±522	2245±289	2146±314	2090±601	0.47
% Carbohydrate	50±10	50±7	48±6	49±8	0.10
% Fat	35±9	35±7	36±5	34±6	0.06
% Saturated fat	38±10	37±7	38±6	41±7	0.93
% Protein	12±4	12±6	15±6	17±3	0.23
Physical activity (steps/day)	6430±1301	6502±834	6416±1131	6305±1385	0.55
Time standing (min/day)	188±119	169±86	194±109	207±95	0.09
Time in supine rest (min/day)	500±190	476±176	491±201	517±190	0.41

	PMEX (n=59)				p-value
	Baseline	4 weeks	8 weeks	16 weeks	
Pre-exercise calorie intake (kcal)	828±48	830±30	829±28	825±40	0.41
24-h calorie intake (kcal/day)	2307±445	2319±557	2299±591	2292±440	0.76
% Carbohydrate	44±7	48±5	48±7	50±8	0.08
% Fat	33±6	31±5	33±9	29±4	0.11
% Saturated fat	38±3	38±9	36±8	38±7	0.44
% Protein	23±5	21±7	18±7	21±3	0.07
Physical activity (steps/day)	5898±1275	6014±1411	5988±1332	6109±1545	0.27
Time standing (min/day)	203±113	199±117	200±187	197±100	0.51
Time in supine rest (min/day)	524±167	523±201	499±192	516±156	0.12

	CONTROL (n=38)				p-value
	Baseline	4 weeks	8 weeks	16 weeks	
Pre-exercise calorie intake (kcal)	-----	-----	-----	-----	
24-h calorie intake (kcal/day)	2207±495	2217±437	2099±559	2073±530	0.18
% Carbohydrate	46±5	48±6	47±9	49±7	0.08
% Fat	35±3	32±5	32±6	33±4	0.19
% Saturated fat	40±8	33±6	36±8	41±6	0.21
% Protein	19±2	20±5	20±9	18±3	0.17
Physical activity (steps/day)	5430± 313	6009±1423	6106±1276	6005±1457	0.07
Time standing (min/day)	195±88	199±98	187±203	207±101	0.29
Time in supine rest (min/day)	488±201	503±158	493±189	492±194	0.52

Now in the RESULTS section, it is explained in more detail: “We could not detect significant changes among groups in 24-h calorie intake, macronutrient distribution, or physical activity **month by month or as reported** before or after 16 weeks of training (all $p > 0.05$ for time x group; Table 4)”.

Participants were required to consume a meal at least 1 h before the start of each training session. Was this confirmed by any way? is it likely that AMEX exercised in fasting conditions?

RESPONSE: Monthly, during the intervention period, (16 weeks), subjects filled out a 3-d nutritional diary recording all foodstuff ingested and the time of ingestion. In the nutritional diary, no subject reported performing the exercise in a fasting state.

Exercise training: 4 x 4-minute intervals at 90% of HRMAX interspersed with a 3-minute active recovery at 70% of HRMAX. Was there any training adaptation? this exercise load seem too hard as a starting point for a sedentary person. Please, describe.

RESPONSE: Subjects were required to hold a workload that elicited 90% of their individual HRmax for 4 minutes repeated 4 times. We followed the progression principle of training and during the first 3 weeks (9 training sessions) the hard bouts were expanded from 2 to 4 min in duration and from 2 bouts to 4 bouts in number. It should be mentioned that 90 % of HRmax can be attained with a workload that is significantly lower than when pedaling at 90% of VO₂max or 90% of Wmax. Participants were 55 years old on average and their HRmax was 162 beats per minute. Thus, on average their 90% of HRmax is 146 beats per minute (see Table 2) and all of them were able to reach their predicted 90% of HRmax three weeks into the training program.

We observed the classical training adaptation response with 11-16% increase in VO₂max, 15-21% increase in Wmax, and 28-29% increase in maximal fat oxidation (bottom of Table 3).

Please describe how was HRMAX reevaluated.

RESPONSE: Monthly, during the Wednesday workout, on the third high-intensity bout of HIIT, participants were required to perform one maximal bout to exhaustion while HR was recorded.

Now in the METHODS section: “Monthly HR_{MAX} was reevaluated during a maximal cycling bout in a regular training session and training workloads adjusted accordingly”.

Blood sampling. When was blood samples taken? how many days after the last exercise session? please specify.

RESPONSE: Thanks for the comment, now we further detail blood sampling.

Now in the METHODS section: “**MetS components and other health indicators.** Before and after the 4 months of intervention patients arrived at the laboratory in the morning after 10–12 h overnight fast. For the training groups, post-training measurements were scheduled at least 48 hours after the last exercise training session to examine the chronic effects of exercise training rather than the acute most recent exercise session. Nude body weight (Hawk; Metler, Toledo, OH), height (Stadiometer; Secca 217, Hamburg, Germany), and waist circumference were measured using a non-elastic measuring tape. Fat mass (FM) and fat-free mass (FFM)

were determined by bioelectrical impedance analysis (Tanita BC-418; Tanita Corp, Tokyo, Japan). After 10 minutes of supine rest, blood pressure was measured in triplicate using a calibrated ECG-gated electro-sphygmomanometer (Tango, Suntec Medical; NC).

Thereafter, **on a different day, within the same week**, a 7-mL blood sample was collected to determine serum glucose, insulin, and lipid levels (triglycerides, total cholesterol, HDL, and LDL-cholesterol). Insulin sensitivity was calculated using the homeostasis model assessment for insulin resistance (HOMA-IR (Matthews *et al.*, 1985)).

The equation to determine MetS Z-Score included MAP instead of the single values of SPB or DPB as indicated in the definitions. Can the authors specify why? please, provide a rationale on this.

RESPONSE: We agree with the reviewer that the equation for determining the MetS Z-Score is more accurate including both SPB and DBP.

We have performed the MetS Z-Score analysis applying the modifications proposed by the reviewer and although the changes are minimal, we have modified (highlighted in red throughout the manuscript) the data in the ABSTRACT, METHODS, RESULTS, DISCUSSION, and FIG 2 sections.

Statistical analysis

There is need to include a detailed description of the main outcome variable and statistical power calculations.

RESPONSE: We estimated the necessary sample size based on changes in Z score from a published study on similar metabolic syndrome patients (PMID: 31415443) who underwent 16 weeks of HIIT training (similar to this intervention). The following website (<https://www.statulator.com/SampleSize/ss2PM.html>) was used to calculate the sample size. The software was set to compare paired differences in Z score from the average difference between before and after training (0.16), and the SD of the differences (0.29).

The statistical power (sample size) calculation is now described in the METHODS: **“Sample size calculation was based on changes in MetS Z score data (main outcome) in individuals with MetS completing a 16 weeks exercise training program similar to the one in this study (Morales-Palomo *et al.*, 2019). Assuming a power of 80% and a α -error probability of 0.05 it was calculated that 28 patients were required to detect a significant effect of exercise training on improving MetS Z score. In that study, there was a 40% improvement in Z score with training. However, the present study is not geared to detect the effects of a training program *per se*, but the more subtle differences between a.m. and p.m. training groups. Since we expected small differences between AMEX. and PMEX groups, we doubled the calculated sample size.”**

The estimations of the effect size are not clear. Please, indicate the main outcome variable, the expected changed and the comparisons you intend to conduct.

RESPONSE: Now in the METHODS section effect size is explained in more detail: “To improve the interpretation of the differences, the effect size of time and time x group interaction were calculated using partial η^2 . The effect size obtained from η^2 was considered large if ≥ 0.14 , moderate ≥ 0.06 , and small if < 0.06 ”.

The authors should control for multiple comparisons. As presented, the authors make comparisons between control vs exercise (including both exercise groups with no rationale to do so), control vs morning, control vs evening, evening vs morning. Please, indicate whether there is enough power to do so.

RESPONSE: The linear mixed model ANCOVA prevented multiple comparisons when not enough statistical power was reached. For each variable, the effects of time (PRE-POST) and the interaction between time effect and each of the experimental groups (time x group interaction) were analyzed. Pairwise comparisons were made between the 3 groups at the two time points only when there was a statistically significant interaction.

Now in the METHODS section, we explain in more detail: “Mixed-design (Split-plot) ANCOVA was run to analyze differences across time (baseline vs 16 weeks of training) and between experimental groups in all reported variables, adjusted by baseline values. This design tested the differences between the 3 groups (AMEX, PMEX, and CONTROL) whilst participants underwent repeated measures (PRE and POST) in the primary and secondary outcome measures. To minimize the risk of statistical type I error, post hoc pairwise comparisons (Tukey test) were only performed between groups when a significant time x group interaction was found”.

The sample combines both men and women, but they may respond differently to the exercise intervention.

RESPONSE: The percentage of women participating in each group was similar (AMEX 38%; PMEX 36% and CONTROL 32%; $p=0.17$) and all women were postmenopausal without hormonal treatment. Thus, it is unlikely that a sex factor could be altering the reported a.m. vs. p.m. responses. Further, we have recently reported (PMID: 31821339) that the metabolic and cardiovascular responses to 4 months of HIIT in MetS patients (without controlling for “time-of-day”) are similar in men and women.

Below is a statistical analysis by sex of the main responses from the current study.

	Women (n=49)		Men (n=90)		p-value	
	Pre	Post	Pre	Post	Time	Time x Sex
Body weight (Kg)	75.6 ± 8.4	75.3 ± 8.6	90.3 ± 10.5	89.2 ± 10.4	<0.001	0.061
MetS Z-score	0.57 ± 0.68	0.50 ± 0.66	0.53 ± 0.55	0.39 ± 0.49	<0.001	0.201
VO ₂ MAX (L·min ⁻¹)	1.49 ± 0.27	1.66 ± 0.30	2.35 ± 0.48	2.53 ± 0.57	<0.001	0.696

VO ₂ MAX (mL·kg ⁻¹ ·min ⁻¹)	19.9 ± 3.6	22.2 ± 4.4	26.3 ± 5.1	28.5 ± 5.7	<0.001	0.679
--	------------	------------	------------	------------	--------	-------

Now in the RESULTS section, we have also included the p-values of the interaction time by sex of the main variables of the study: “Women participants comprised 35% of all subjects. Data were analyzed without sex differentiation since all women were postmenopausal and were not taking hormonal replacement (Guio de Prada et al., 2019), and their responses did not differ from men’s responses **in primary and secondary outcomes of the study (time x sex interaction; MetS Z score, p=0.20; body weight (Kg), p=0.06); VO₂MAX (L·min⁻¹), p=0.70)**”

It would be appropriate to conduct linear mixed models.

RESPONSE: We agree with the reviewer and now it is better explained in the METHODS section that we used a linear mixed model: “Mixed-design (Split-plot) ANCOVA was run to analyze differences across time (baseline vs 16 weeks of training) and between experimental groups in all reported variables, adjusted by baseline values. **This design tested the differences between the 3 groups (AMEX, PMEX, and CONTROL) whilst participants underwent repeated measures (PRE and POST) in the primary and secondary outcome measures. To minimize the risk of statistical type I error, post hoc pairwise comparisons (Tukey test) were only performed between groups when a significant time x group interaction was found**”.

Include details on how was sex treated.

RESPONSE: Already answered in the previous point

Please, describe the statistical models more in detailed. Dependent variables, etc. Was the changed included in the model? the post-intervention value?

RESPONSE: Thanks to your correct previous comments, we have expanded the description of our statistical design and outcomes in the following sections:

INTRODUCTION: ... “The purpose of this study was to determine if there are differences in the therapeutic impact of a 16-week-long high-intensity aerobic exercise program **on MetS Z Score (primary outcome)**, performed either in the morning or **afternoon**”.

METHODS: ...”Mixed-design (Split-plot) ANCOVA was run to analyze differences across time (baseline vs. 16 weeks of training) and between experimental groups in all reported variables, adjusted by baseline values. **This design tested the differences between the 3 groups (AMEX, PMEX, and CONTROL) whilst participants underwent repeated measures (PRE and POST) in the primary and secondary outcome measures. To minimize the risk of statistical type I error, post hoc pairwise comparisons (Tukey test) were only performed between groups when a significant time x group interaction was found**”.

RESULTS: ...”Participants were all white living in southern Europe. Women participants comprised 35% of all subjects. Data were analyzed without sex differentiation since all women were postmenopausal and were not taking hormonal replacement (Guio de Prada *et al.*, 2019), and their responses did not differ from men’s responses **in primary and secondary outcomes of the study (time x sex interaction; MetS Z score, p=0.20; body weight (Kg), p=0.06); VO₂MAX (L·min⁻¹), p=0.70, respectively)**”.

Covariates. only the baseline values was included? but if baseline levels were similar across groups, is this correct to do it?

RESPONSE: Despite that ANOVA did not detect baseline differences (before training) among groups, the assumption of equal variance is critical to the ANOVA. Given the unequal sample variance, we thought that covariate each variable with its baseline values ensured confidence in treatment differential effects. When analyzing the effects of the baseline values on the changes in MetS Z score, we found that there was a significant covariance that was not found in other variables (sex, age, and weight).

Information about how were missing values treated should be included.

RESPONSE: We used per-protocol analysis, including only those patients who completed the treatment in the statistical analysis.

Sample loss (attrition) was relatively low after intervention and well-balanced between groups (P=0.59; Table 2). Using "intention to treat" analysis, entails predicting their responses using extrapolation from baseline to 16 weeks of data, and that introduces a more significant bias than the per-protocol analysis we used.

Results

The authors refer to another study to indicate that women responses did not differ from men's responses? what about in the current study? the authors should provide data by sex in the current study at least in supplementary file and conduct formal sex-interaction analyses.

RESPONSE: Already answered in the table above.

Table 1 and 2. The P value is from the t test or from the chi-square, right? I believe the authors are not providing the chi-square but the P value from the chi-square test. Please, indicate more clearly. Also include its description in the statistical analysis. Is randomisation is correct, to compared baseline values among groups is not needed nor correct

RESPONSE: We agree with the reviewer that the p-value expressed in Tables 1 and 3 should be better described. The values expressed in both Tables 1 and 2 are p values obtained by one-way ANOVA (Table 1), student t-test for independent samples (Table 2), and chi-square test.

Now in the METHODS section: "Between-group comparisons at baseline were performed with a one-way ANOVA (Table 1) and student t-test for independent samples (Table 2). When categorical variables were analyzed, the Chi-square test was performed. (Table 1)".

RESULTS section "The withdrawal rate was similar in all three groups (AMEX,16%; PMEX; 21% and CONTROL 24%, p-value of Chi-square=0.59) and always took place during the first 3 weeks of the experiment. Discontinuation in study participation obeyed to changes in work schedule, unable to attend to the post-16-week test, and minor injuries or diseases (back pain and prolonged flu-like symptoms; Figure 1)".

In addition, Tables 1 and 2 have been modified to better show the p-value or the p-value of chi-square.

Regarding comparisons of the baseline values, we believe that although the randomization was correct, the data offers valuable information for the reader.

Can the authors specify if the Avg heart rate is from the session?

RESPONSE: Yes, it is the average of the whole training program for each experimental group. Following the reviewer's recommendation, we have now included in Table 2 the average heart rate of the HIIT intervals.

Table 2. Characteristics of training bouts.

	AMEX (N =42)	PMEX (N =59)	CONTROL (N =38)	p-value
Training sessions compliance	92%	90%	-	-
Exercise intensity				
Avg heart rate (bt·min ⁻¹)	130±12	133±15	-	0.23
Avg heart rate at 70% (bt·min ⁻¹)	116±13	119±15	-	0.37
Avg heart rate at 90% (bt·min ⁻¹)	145±15	147±20	-	0.28
Avg heart rate (%HR _{MAX})	79%	81%	-	0.56
Avg heart rate at 70% (%HR _{MAX})	71%	72%	-	0.88
Avg heart rate at 90% (%HR _{MAX})	90%	92%	-	0.47
Avg workload (W)	114±29	118±26	-	0.25
Avg workload (%W _{MAX})	59%	56%	-	0.43
Avg $\dot{V}O_2$ (L·min ⁻¹)	1.33±0.39	1.41±0.42	-	0.36
Avg $\dot{V}O_2$ (% $\dot{V}O_{2MAX}$)	69%	67%	-	0.42
Avg energy expenditure (Kcal)	383±80	393±85	-	0.52

Data presented as mean ± SD

Can the authors provide the Avg heart rate of the HIIT intervals? was the 90% achieved always?

RESPONSE: During all training sessions, the heart rate of each participant was continuously displayed on a large screen using training monitoring software (Seego Realtrack Systems, Spain). This system makes each session very motivating for the participants since they adjust the workload to “enter” their “zone”. Every single training session was recorded and supervised by a member of our team to ensure that all individuals reached their target HR.

Was VO2 measured in every exercise session? please specify and describe in methods

RESPONSE: Thanks for the comment. We agree that we should have described how the VO₂ and energy expenditure data were obtained.

Now in the METHODS section it is described in detail: “Mid-training (week 8) all subjects in each exercising group were studied in the laboratory during a regular exercise training session. VO₂, VCO₂, heart rate, and power output were collected continuously during the whole exercise session. VO₂ and VCO₂ were used to calculate energy expenditure as proposed by Brouwer (Brouwer, 1957)”.

Please, describe in detail how was energy expenditure determined.

RESPONSE: Please see the previous point

Discussion

The summary of the first paragraph of the discussion is not fully correct, as no all CVD risk markers differ between AMEX and PMEX.

RESPONSE: We agree with the reviewer that not all CVC risk markers differ between exercise groups and therefore the discussion summary sentence needs to be rephrased.

Now in the DISCUSSION section the sentence has been rewritten: “In summary, our data suggest that exercise training for 16 weeks in the morning (AMEX) is more efficient at reducing insulin sensitivity (i.e., insulin concentration and insulin resistance (HOMA-IR)) and systolic blood pressure than similar training in the afternoon (PMEX)”.

Please, discuss about the clinical meaning of the differences observed between AMEX and PMEX. Do the authors genuinely hold the belief that the observed "minor" changes hold clinical significance substantial enough to warrant recommending morning exercise for individuals with MetS?

RESPONSE: We agree with you that the health benefits derived from the exercise program in both groups regardless of time of day should be highlighted.

We have made several changes to the manuscript to convey this idea in the KEY POINTS, ABSTRACT and DISCUSSION sections:

KEY POINTS:

- The effect of exercise time of day on health promotion is an area that has gained interest in recent years however; large-scale, randomized-control studies are scarce.
- People with metabolic syndrome are at risk of developing cardiometabolic diseases and reductions in this risk with exercise training can be precisely gauged using a compound score sensitive to subtle evolution in each MetS component (i.e., Z score).
- Supervised aerobic exercise for 16 weeks (morning and afternoon), without dietary restriction, improved cardiorespiratory and metabolic fitness, body composition, and mean arterial pressure when compared to a non-exercise control group.

- **However**, training in the morning, without changes in exercise dose or intensity, reduces **systolic** blood pressure, insulin resistance, and metabolic syndrome Z score, further than when training in the afternoon.
- Thus, exercise training in the morning **somewhat** more efficiently improves cardiovascular and metabolic health in individuals with metabolic syndrome.

ABSTRACT

“In summary, exercise training in the morning in comparison to training in the afternoon is **somewhat** more efficient at reducing the cardiometabolic risk factors that compose the metabolic syndrome”.

DISCUSSION

“We corroborated that 16 weeks of supervised high-intensity interval aerobic exercise (**AMEX and PMEX**), without dietary restriction, improved cardiorespiratory and metabolic fitness and body composition when compared to a non-exercise **CONTROL** group. **The findings support the concept that exercise training, regardless of the time of day, improves the fitness and health of individuals with metabolic MetS.** The novel finding was that systolic blood pressure, MetS Z score, blood insulin concentration, and insulin resistance were improved further in the morning than in the afternoon training group (AMEX vs PMEX). Our data suggest that morning training is a **somewhat** more effective strategy to maximize health promotion in individuals with MetS”.

Dear Dr Mora-Rodriguez,

Re: JP-RP-2023-285366R1 "Efficacy of morning versus afternoon exercise training on reducing metabolic syndrome components: A randomized controlled trial" by Felix Morales-Palomo, Alfonso Moreno-Cabañas, Laura Alvarez-Jimenez, Diego Mora-Gonzalez, Juan Fernando Ortega, and Ricardo Mora-Rodriguez

Thank you for submitting your manuscript to The Journal of Physiology. It has been assessed by a Reviewing Editor and by 2 expert referees and we are pleased to tell you that it is acceptable for publication following satisfactory revision.

REVISION CHECKLIST:

Please upload two versions of your manuscript text: one with all relevant changes highlighted and one clean version with no changes tracked. The manuscript file should include all tables and figure legends, but each figure/graph should be uploaded as separate, high-resolution files. The journal is now integrated with Wiley's Image Checking service. For further details, see: <https://www.wiley.com/en-us/network/publishing/research-publishing/trending-stories/upholding-image-integrity-wileys-image-screening-service>.

We look forward to receiving your revised submission.

Yours sincerely,

Paul Greenhaff
Senior Editor
The Journal of Physiology

REQUIRED ITEMS

- The Journal of Physiology funds authors of provisionally accepted papers to use the premium BioRender site to create high resolution schematic figures. Follow this link and enter your details and the manuscript number to create and download figures. Upload these as the figure files for your revised submission. If you choose not to take up this offer we require figures to be of similar quality and resolution. If you are opting out of this service to authors, state this in the Comments section on the Detailed Information page of the submission form. The link provided should only be used for the purposes of this submission. Authors will be charged for figures created on this premium BioRender account if they are not related to this manuscript submission.

- Papers must comply with the Statistics Policy: https://jp.msubmit.net/cgi-bin/main.plex?form_type=display_requirements#statistics.

In summary:

- If $n \leq 30$, all data points must be plotted in the figure in a way that reveals their range and distribution. A bar graph with data points overlaid, a box and whisker plot or a violin plot (preferably with data points included) are acceptable formats.

- If $n > 30$, then the entire raw dataset must be made available either as supporting information, or hosted on a not-for-profit repository e.g. FigShare, with access details provided in the manuscript.

- 'n' clearly defined (e.g. x cells from y slices in z animals) in the Methods. Authors should be mindful of pseudoreplication.

- All relevant 'n' values must be clearly stated in the main text, figures and tables.

- The most appropriate summary statistic (e.g. mean or median and standard deviation) must be used. Standard Error of the Mean (SEM) alone is not permitted, unless justified and presented alongside confidence intervals.

- Exact p values must be stated. Authors must not use 'greater than' or 'less than'. Exact p values must be stated to three significant figures even when 'no statistical significance' is claimed.

EDITOR COMMENTS

Reviewing Editor:

Reviewer 2 made some additional suggestions that should be considered by the authors.

Senior Editor:

Thank you for revising the manuscript, which has been reviewed by the same RE and referees that considered the previous version. All feel the manuscript has been improved. However, Reviewer 2 raises a number of further points that need attention. Most relate to sufficient information being included in the manuscript to allow clarity of understanding by the reader. However, one point focused on whether the sample size of the study has been sufficiently powered is very important. Indeed, it is crucial the authors fully address this point going forward as it is key to the conclusions being made in the manuscript. Just to be clear, I do not concur with the view of Reviewer 2 that it should be ok to mention that this is a pilot study or a preliminary study. If one study group has indeed been underpowered this will of course have a major impact on data interpretation and therefore the conclusions presented, which would need to be moderated accordingly.

REFeree COMMENTS

Referee #1:

Thanks for addressing my comments.

Referee #2:

The manuscript has improved considerably, but from my point of view, there are still there a number of points that need to be carefully considered.

The title says: Efficacy of morning versus afternoon exercise training on reducing metabolic syndrome components: A randomized controlled trial.

In order to make it more accurate, the title should reflect the type of exercise performed, as exercise is a very general term. This should be specified all through the text when referring to the 'exercise program' performed in this study. The authors have referred some times to "high-intensity aerobic exercise program".

The authors mention: However, training in the morning, without changes in exercise dose or intensity, reduces systolic blood pressure, insulin resistance, and metabolic syndrome Z score, further than when training in the afternoon.

Metabolic syndrome Z score is reduced as a consequence of a reduction of systolic blood pressure, so I believe it is redundant to mention.

Abstract

There are a number of abbreviations that are not explained and makes the reading difficult. E.g. FO.

In the summary, the authors mention: In summary, exercise training in the morning in comparison to training in the afternoon is somewhat more efficient at reducing the cardiometabolic risk factors that compose the metabolic syndrome. Actually, the 'only' cardiometabolic risk factors that compose the metabolic syndrome that changes after the intervention in the morning vs afternoon is systolic blood pressure. Please, be more specific.

Introduction

The authors mention in the last sentences of the second paragraph that synchronizing exercise sessions with an individual's circadian rhythms could prove to be an effective approach for maximizing the positive health outcomes derived from physical activity. Have the authors collected information about the individual's circadian rhythms? If not, this should be acknowledged in the limitations section. If this info has been collected, it should be mentioned and discussed appropriately.

Methods

Since subjects were under medical treatment, please, include in the table 3 or 4 the changes in medications after the intervention. The authors indicate that there was no changes in the use of medications after the intervention in any group, but these data are not presented anywhere.

End of first paragraph of methods, please, specify the effects of XXX on what? By the way, investigating "the effects of interactions of medicine and exercise with meal timing" seems complex. We encounter a multifaceted area of study that can initially appear intricate to comprehend. Can you please be a bit more descriptive?

Please include in detailed the exercise progression in the manuscript, so that the reader can understand what was actually done.

Please, provide details on how VO₂, VCO₂, heart rate, and power output were collected continuously during the whole exercise session, i.e. details of the equipment, time frame, etc so that the methods can be replicable.

Statistical power

The authors have made estimations on sample size calculation based on a previous study of similar characteristics and have come up with 28 participants per group to detect significant changes on MetS Z score before-after an exercise intervention. Since this study intended to detect changes between morning and afternoon exercise, they have doubled the sample size without actually knowing whether this is enough. Moreover, despite they say have doubled the number of people required (n=28) looking at the number of people included in this study, only the afternoon group meets with this criterium. This indicates that just following the author's estimations (which need to be confirmed are correct), the sample size is not enough to have enough statistical power. This is crucial and should be taking into consideration in detail. If this is

a substudy of a larger study, it should be ok to mention that this is a pilot study or a preliminary study as mentioned in the response letter, but the authors should be clear since the mistake is very evident in the calculations done.

Sex dimension. I believe the sex dimension has not been considered or mentioned in the manuscript, and this is a topic of recent interest that should be carefully considered. The authors have presented, in the response letter, the sex*time interaction, but this study includes the time dimension which should be taken into account in the sex dimension. The fact that all women are postmenopausal does not guarantee that their response in the morning exercise is equal to men.

Since PA and diet was monitored during the intervention (monthly), the authors can provide the tables shown in the response letter as supplementary tables.

Discussion

Please, include the population type at the end of the last sentence in the first paragraph of the discussion.

Table 3, second block of variables, include systolic and diastolic blood pressure together with the other MetS parameters and change the mean arterial pressure to the next block.

Include also the mean and SD values of Z score at baseline and 16wk

Table 4. please include also the changes in drugs all through the exercise intervention.

Figure 1, the word evening is employed and should be changed.

Figure - abstract. Please include 8-9h to comply with what it is mentioned in the methods.

Please refer to waist circumference all through the manuscript. Some times, it is used waist perimeter.

END OF COMMENTS

1st Confidential Review

28-Aug-2023

RESPONSES TO REVIEWERS

We appreciate the time devoted by the reviewers to comment on the manuscript. After incorporating all the corrections, we believe the manuscript has improved notably. Please find below an itemized response to your comments. Parallel changes in the manuscript are highlighted in red.

Referee #1:

Thanks for addressing my comments.

Referee #2:

The manuscript has improved considerably, but from my point of view, there are still there a number of points that need to be carefully considered.

The title says: Efficacy of morning versus afternoon exercise training on reducing metabolic syndrome components: A randomized controlled trial.

In order to make it more accurate, the title should reflect the type of exercise performed, as exercise is a very general term. This should be specified all through the text when referring to the 'exercise program' performed in this study. The authors have referred some times to "high-intensity aerobic exercise program".

RESPONSE: We agree with reviewer. Now the TITLE has been changed to: "Efficacy of morning versus afternoon **aerobic** exercise training on reducing metabolic syndrome components: A randomized controlled trial".

According to Gibala's classification high-intensity interval training, sprint interval training, and moderately intense continuous training can all be considered aerobic training (PMID: 27748956). Throughout the manuscript, we specify "high-intensity aerobic exercise training program".

The authors mention: However, training in the morning, without changes in exercise dose or intensity, reduces systolic blood pressure, insulin resistance, and metabolic syndrome Z score, further than when training in the afternoon.

Metabolic syndrome Z score is reduced as a consequence of a reduction of systolic blood pressure, so I believe it is redundant to mention.

RESPONSE: We agree with the reviewer that mentioning MetS may be redundant. Now, we have modified KEY POINTS and CONCLUSIONS to avoid that redundancy.

Abstract

There are a number of abbreviations that are not explained and makes the reading difficult. E.g. FO.

RESPONSE: Now highlighted in red in the ABSTRACT, we spelled out the following: "Exercise training was comprised of 48 supervised high-intensity interval sessions distributed over 16 weeks. Body composition, cardiorespiratory **fitness (assessed by $\dot{V}O_{2MAX}$)**, **maximal fat oxidation (FO_{MAX})**, blood pressure, and blood metabolites were assessed before and after the intervention".

In the summary, the authors mention: In summary, exercise training in the morning in comparison to training in the afternoon is somewhat more efficient at reducing the cardiometabolic risk factors that compose the metabolic syndrome. Actually, the 'only' cardiometabolic risk factors that compose the metabolic syndrome that changes after the intervention in the morning vs afternoon is systolic blood pressure. Please, be more specific.

RESPONSE: Now the summary sentence of the ABSTRACT is more specific: “In summary, **high-intensity aerobic** exercise training in the morning in comparison to training in the afternoon is somewhat more efficient at reducing cardiometabolic risk factors (i.e., **systolic blood pressure and insulin sensitivity**).”.

Introduction

The authors mention in the last sentences of the second paragraph that synchronizing exercise sessions with an individual's circadian rhythms could prove to be an effective approach for maximizing the positive health outcomes derived from physical activity. Have the authors collected information about the individual's circadian rhythms? If not, this should be acknowledged in the limitations section. If this info has been collected, it should be mentioned and discussed appropriately.

RESPONSE: We agree that synchronizing exercise sessions with circadian rhythms could be an effective approach to maximize health outcomes. However, surveys trying to detect individual's chronotype were not used in this study and that is now acknowledged in the limitation section of the discussion: “**We did not assess participants' circadian preference for the timing of various activities (i.e., morningness vs. eveningness questionnaire). We hope that subject randomization resulted in an even distribution of those individual types in the three experimental groups.**”

Methods

Since subjects were under medical treatment, please, include in the table 3 or 4 the changes in medications after the intervention. The authors indicate that there was no changes in the use of medications after the intervention in any group, but these data are not presented anywhere.

RESPONSE: The changes in medication are now better described in the RESULTS: “There were also no changes in the use of medications **to treat MetS** after the intervention in any group (**p=0.37 for time x group; Table 3**)”.

Also, the number of medications prescribed for the treatment of metabolic syndrome has been included in Table 3, in the three groups at baseline and after the 16 weeks of training. There was not a time effect or a time per group interaction for the use of medicines.

End of first paragraph of methods, please, specify the effects of XXX on what? By the way, investigating "the effects of interactions of medicine and exercise with meal timing" seems complex. We encounter a multifaceted area of study that can initially appear intricate to comprehend. Can you please be a bit more descriptive?

RESPONSE: That sentence now reads:

“This is a substudy part of a larger clinical trial evaluating **the interactions** between the **timing of oral medicines, exercise training, and meal intake in the evolution of the factors that compose** the MetS (ClinicalTrials.gov Identifier: NCT04477590).”

Please include in detailed the exercise progression in the manuscript, so that the reader can understand what was actually done.

RESPONSE: Thanks for the comment, now we further detail the exercise progression.

Now in the METHODS section, we included:

“We followed the progression principle of training and during the first 3 weeks (9 training sessions) the hard bouts were expanded from 2 to 4 min in duration and from 2 bouts to 4 bouts in number”.

Please, provide details on how VO₂, VCO₂, heart rate, and power output were collected continuously during the whole exercise session, i.e. details of the equipment, time frame, etc so that the methods can be replicable.

RESPONSE: Thanks for the comment. We agree that we should have described how the VO₂ and energy expenditure data were obtained.

Now in the METHODS section, it is described in detail as follows:

“Mid-training (week 8) all subjects in each exercising group were studied in the laboratory during a regular exercise high-intensity interval session using an electronically braked cycle ergometer (Ergoselect 200; Ergoline, Germany). VO₂, VCO₂, heart rate, and power output were collected continuously during the whole exercise session. Exhaled air was continuously collected and analyzed breath-by-breath for oxygen consumption and carbon dioxide production using indirect calorimetry (Quark b2; Cosmed, Italy). VO₂ and VCO₂ were used to calculate energy expenditure as proposed by Brouwer (Brouwer, 1957)”.

Statistical power

The authors have made estimations on sample size calculation based on a previous study of similar characteristics and have come up with 28 participants per group to detect significant changes on MetS Z score before-after an exercise intervention. Since this study intended to detect changes between morning and afternoon exercise, they have doubled the sample size without actually knowing whether this is enough. Moreover, despite they say have doubled the number of people required (n=28) looking at the number of people included in this study, only the afternoon group meets with this criterium. This indicates that just following the author's estimations (which need to be confirmed are correct), the sample size is not enough to have enough statistical power. This is crucial and should be taking into consideration in detail. If this is a substudy of a larger study, it should be ok to mention that this is a pilot study or a preliminary study as mentioned in the response letter, but the authors should be clear since the mistake is very evident in the calculations done.

RESPONSE: We have conducted a deeper search in the literature reporting MetS Z score changes and conducted a more complete sample size analysis, to avoid what it looked like “guessing” in our initial estimations for sample size. We (Mora 2016, 2018, Morales 2019) and others (Johnson 2007, Earnest 2013) have reported improvements in MetS Z-score with exercise training of similar characteristics (duration and intensity) to the one proposed in this study. The changes in MetS Z-score after 4-6 months of exercise compared to the control group, ranged between 0.229 with a pooled standard deviation of 0.267. Assuming similar

differences between treatments, the present study would require a sample size of 22 individuals for each group to achieve a power of 80% and a level of significance of 5% (two-sided), for detecting a true difference in means (Dhand, N. K., & Khatkar, M. S. (2014). Statulator: An online statistical calculator. Sample Size Calculator for Comparing Two Independent Means. Accessed 28 September 2023 at <http://statulator.com/SampleSize/ss2M.html>). Because the attrition rates in exercise interventions could reach 20% (PMID: 19951417 and PMID: 36502286) and because it is uncertain what the differences would be between training in the AM vs PM, we doubled the calculated required sample size to a target sample of at least 50 participants in each group.

The table below shows results from the effects of aerobic training in MetS Z-score in 6 studies (5 published and 1 unpublished) in middle-aged individuals with metabolic syndrome of similar characteristics to the sample under study.

PUBLICATION	Z score results	Treatment (sample)	Changes in Z score \pm SD	Sample size needed (80% power 5% significance)	Effect size of difference between groups
Johanna L. Johnson, et al. Exercise Training Amount and Intensity Effects on Metabolic Syndrome (From Studies of a Targeted Risk Reduction Intervention through Defined Exercise). Am J Cardiol. 2007 December 15; 100(12): 1759–1766. (* values divided by 5 MetS factors to comply with units in the rest of the studies)	Table 3 and figure 2, pag, 1773	6 months of vigorous intensity aerobic training (n=44; >1500 kcals per week) compared to a control group (n=41).	Exercise group -0.28 \pm 0.34 Control group +0.0 \pm 0.30 Difference between groups -0.28 \pm 0.32	21 per group	0.875
Earnest CP et al. Dose effect of cardiorespiratory exercise on metabolic syndrome in postmenopausal women. Am J Cardiol. 2013 ;111(12):1805–11.	Figure 1, pg 9	6 months of continuous pedaling at 50% VO ₂ max (~996 kcals week) in post menopausal women (n=103) compared to a control group (n=155).	Exercise group -0.13 \pm 0.07 Control group -0.023 \pm 0.07 Difference between groups -0.107 \pm 0.07	8 per group	1.46
R. Mora-Rodriguez, et al., Effects of Simultaneous or Sequential Weight Loss Diet and Aerobic Interval Training on Metabolic Syndrome. Int J Sports Med. 2016	Figure 2, pg 278	4 months of high intensity aerobic interval training (n=36), 3 times per week (~900 kcals week) + 4 months of diet.	Exercise group -0.36 \pm 0.173 Control group -0.02 \pm 0.277 Difference between groups -0.34 \pm 0.225	7 per group	1.51

Apr;37(4):274-8. (* values divided by 5 MetS factors to comply with units in the rest of the studies)					
	Table 1, pg 1269	4 months of high intensity aerobic interval training (n=138), 3 times per week (~900 kcal week) compared to a control group (n=22).	Exercise group -0.21 ± 0.45 Control group +0.03 ± 0.45 Difference between groups -0.24 ± 0.45	56 per group	0.533
F. Morales-Palomo, et al. Effectiveness of Aerobic Exercise Programs for Health Promotion in Metabolic Syndrome. Med. Sci. Sports Exerc, Vol. 51, No. 9, pp. 1876–1883, 2019.	Figure 3A, pg 1881	4 months of high intensity aerobic interval training (n=32), 3 times per week, compared to a control group (n=22)	Exercise group -0.16 ± 0.29 Control group +0.07 ± 0.22 Difference between groups -0.23 ± 0.255	20 per group	0.902
Unpublished Data base Exercise Physiology Lab (2014-2023)	Unpublished	4 months of high intensity aerobic interval training (n=202), 3 times per week, compared to a control group (n=62)	Exercise group -0.20 ± 0.32 Control group -0.02 ± 0.31 Difference between groups -0.18 ± 0.315	49 per group	0.571
AVERAGE VALUES FROM 6 PUBLISHED AND 1 UNPUBLISHED STUDY (total n =)			0.229 ± 0.267	26.8 per group	0.858

Now in the METHODS section, it is described in detail as follows:

“Sample size calculation was based on changes in MetS Z score data (main outcome) in individuals with MetS completing 4-6 months of aerobic exercise training program similar to the one in this study (Johnson *et al.*, 2007; Earnest *et al.*, 2013; Mora-Rodriguez *et al.*, 2016; Mora-Rodriguez *et al.*, 2018; Morales-Palomo *et al.*, 2019). Assuming a power of 80% and a α -error probability of 0.05 it was calculated that 22 patients were required to detect a significant effect of exercise training on improving MetS Z score. Because the attrition rates in exercise interventions could reach 20% (Groeneveld *et al.*, 2009) and because it is uncertain what the differences would be between training in the AM vs. PM, we doubled the calculated required sample size to a target sample of at least 50 participants in each group”.

Sex dimension. I believe the sex dimension has not been considered or mentioned in the manuscript, and this is a topic of recent interest that should be carefully considered. The authors have presented, in the response letter, the sex*time interaction, but this study includes the time dimension which should be taken into account in the sex dimension. The fact that all women are postmenopausal does not guarantee that their response in the morning exercise is equal to men.

Since PA and diet was monitored during the intervention (monthly), the authors can provide the tables shown in the response letter as supplementary tables.

RESPONSE: Thanks for both comments. With respect to the sex dimension, we agree with the reviewer that despite that our statistics do not show significance for a sex*time interaction (see first paragraph of the results), there may be an influence of sex in the responses that we are not able to detect.

Regarding physical activity and diet monitoring during the 4 months of the study, we now include a table in the supplementary materials that shows those responses.

Now in the RESULTS and SUPPORTING INFORMATION section, it is described like this: “We could not detect significant changes among groups in 24-hour calorie intake, macronutrient distribution, or physical activity month by month (see supporting file) or as reported before or after 16 weeks of training (all $p>0.05$ for time x group; Table 4)”.

Supporting Information

Baseline and monthly evolution of the 24-hour caloric intake and physical activity monitoring in each experimental group

Discussion

Please, include the population type at the end of the last sentence in the first paragraph of the discussion.

RESPONSE: The type of population has been included at the end of the last sentence of the first paragraph of the DISCUSSION: “In summary, our data suggest that **high-intensity aerobic** exercise training for 16 weeks in the morning (AMEX) is more efficient at reducing insulin sensitivity (i.e., insulin concentration and insulin resistance (HOMA-IR)) and systolic blood pressure than similar training in the afternoon (PMEX) **in individuals with MetS**”.

Table 3, second block of variables, include systolic and diastolic blood pressure together with the other MetS parameters and change the mean arterial pressure to the next block.

RESPONSE: We agree with the reviewer. Table 3 has been modified and the second block now includes systolic and diastolic blood pressure instead of mean arterial pressure.

Include also the mean and SD values of Z score at baseline and 16wk

RESPONSE: The Z score values have been included in Table 3.

Table 4. please include also the changes in drugs all through the exercise intervention.

RESPONSE: The number of medications for the treatment of metabolic syndrome has been NOW included in Table 3. Also, the lack of changes in medication are highlighted in the RESULTS: “There were also no changes in the use of medications **to treat MetS** after the intervention in any group ($p=0.37$ for time x group; Table 3)”.

Figure 1, the word evening is employed and should be changed.

RESPONSE: Thanks for this important correction. The word evening in Figure 1 has been changed to afternoon.

Figure - abstract. Please include 8-9h to comply with what it is mentioned in the methods.

RESPONSE: Thanks for warning of that typo. We have included 8-9 h in the figure-abstract.

Please refer to waist circumference all through the manuscript. Some times, it is used waist perimeter.

RESPONSE: Thank you for detecting that mistake. We have now replaced waist perimeter for waist circumference throughout the manuscript.

Dear Dr Mora-Rodriguez,

Re: JP-RP-2023-285366R2 "Efficacy of morning versus afternoon aerobic exercise training on reducing metabolic syndrome components: A randomized controlled trial" by Felix Morales-Palomo, Alfonso Moreno-Cabañas, Laura Alvarez-Jimenez, Diego Mora-Gonzalez, Juan Fernando Ortega, and Ricardo Mora-Rodriguez

We are pleased to tell you that your paper has been accepted for publication in The Journal of Physiology.

Authors should note that it is too late at this point to offer corrections prior to proofing. The accepted version will be published online, ahead of the copy edited and typeset version being made available. Major corrections at proof stage, such as changes to figures, will be referred to the Editors for approval before they can be incorporated. Only minor changes, such as to style and consistency, should be made at proof stage. Changes that need to be made after proof stage will usually require a formal correction notice.

Yours sincerely,

Paul Greenhaff
Senior Editor
The Journal of Physiology

P.S. - You can help your research get the attention it deserves! Check out Wiley's free Promotion Guide for best-practice recommendations for promoting your work at www.wileyauthors.com/eoo/guide. You can learn more about Wiley Editing Services which offers professional video, design, and writing services to create shareable video abstracts, infographics, conference posters, lay summaries, and research news stories for your research at www.wileyauthors.com/eoo/promotion.

IMPORTANT NOTICE ABOUT OPEN ACCESS: To assist authors whose funding agencies mandate public access to published research findings sooner than 12 months after publication, The Journal of Physiology allows authors to pay an Open Access (OA) fee to have their papers made freely available immediately on publication.

You can check if your funder or institution has a Wiley Open Access Account here: <https://authorservices.wiley.com/author-resources/Journal-Authors/licensing-and-open-access/open-access/author-compliance-tool.html>.

EDITOR COMMENTS

Reviewing Editor:

No further comments.

Senior Editor:

Thank you for making the further revisions to the manuscript. It is now acceptable for publication. Congratulations.

REFeree COMMENTS

Referee #1:

Thanks for addressing my comments.

Referee #2:

The authors have made a great job, and the manuscript includes now detailed and accurate information. The findings are of interest and the topic is very timing.

2nd Confidential Review

03-Oct-2023